# Deep brain stimulation of the *Tbr1*-deficient mouse model of autism spectrum disorder at the basolateral amygdala alters amygdalar connectivity, whole-brain synchronization, and social behaviors

**Tsan-Ting Hsu**[1], **Tzyy-Nan Huang**[1], **Chien-Yao Wang**[2], **Yi-Ping Hsueh**[1] *

**1** Institute of Molecular Biology, Academia Sinica, Taipei, Taiwan, Republic of China, **2** Institute of Information Science, Academia Sinica, Taipei, Taiwan, Republic of China

* yph@gate.sinica.edu.tw

## Abstract

Autism spectrum disorders (ASDs) are considered neural dysconnectivity syndromes. To better understand ASD and uncover potential treatments, it is imperative to know and dissect the connectivity deficits under conditions of autism. Here, we apply a whole-brain immunostaining and quantification platform to demonstrate impaired structural and functional connectivity and aberrant whole-brain synchronization in a *Tbr1*⁺/⁻ autism mouse model. We express a channelrhodopsin variant oChIEF fused with Citrine at the basolateral amygdala (BLA) to outline the axonal projections of BLA neurons. By activating the BLA under blue light theta-burst stimulation (TBS), we then evaluate the effect of BLA activation on C-FOS expression at a whole brain level to represent neural activity. We show that *Tbr1* haploinsufficiency almost completely disrupts contralateral BLA axonal projections and results in mistargeting in both ipsilateral and contralateral hemispheres, thereby globally altering BLA functional connectivity. Based on correlated C-FOS expression among brain regions, we further show that *Tbr1* deficiency severely disrupts whole-brain synchronization in the absence of salient stimulation. *Tbr1*⁺/⁻ and wild-type (WT) mice exhibit opposing responses to TBS-induced amygdalar activation, reducing synchronization in WT mice but enhancing it in *Tbr1*⁺/⁻ mice. Whole-brain modular organization and intermodule connectivity are also affected by Tbr1 deficiency and amygdalar activation. Following BLA activation by TBS, the synchronizations of the whole brain and the default mode network, a specific subnetwork highly relevant to ASD, are enhanced in *Tbr1*⁺/⁻ mice, implying a potential ameliorating effect of amygdalar stimulation on brain function. Indeed, TBS-mediated BLA activation increases nose-to-nose social interactions of *Tbr1*⁺/⁻ mice, strengthening evidence for the role of amygdalar connectivity in social behaviors. Our high-resolution analytical platform reveals the inter- and intrahemispheric connectopathies arising from ASD. Our study emphasizes the defective synchronization at a whole-brain scale caused by *Tbr1* deficiency and implies a potential beneficial effect of deep brain stimulation at the amygdala for TBR1-linked autism.

**Data Availability Statement:** All relevant data are within the paper and its Supporting Information files. The file S1 Data contains the numerical value data for Figs 2, 3 and S8. The file S2 Data contains the numerical value data for Fig 4. The file S3 Data contains the numerical value data for Figs 5–7 and S5–S7. The file S4 Data contains the numerical value data for Figs 8B–8C and S9–S10. Custom codes used in the current study have been uploaded to GitHub (https://github.com/HsuehYiPing/Tbr1Amyg) and Zenodo (DOI:10.5281/zenodo.12591119).

**Funding:** This work was supported by grants from Academia Sinica (https://www.sinica.edu.tw, AS-IA-111-L01 and AS-TP-110-L10 to Y.-P.H.), and the National Science and Technology Council (https://www.nstc.gov.tw/?, NSTC 112-2326-B-001-008 to Y.-P.H.). The funders had no role in study design, data collection and analysis, the decision to publish or the preparation of the manuscript.

**Competing interests:** The authors have declared that no competing interests exist.

**Abbreviations:** AAV, adeno-associated virus; ACPR, affected contralateral projection region; AIPR, affected ipsilateral projection region; AP, anterior-posterior; ASD, autism spectrum disorder; BLA, basolateral amygdala; CCFv3, Common Coordinate Framework version 3; ChR, channelrhodopsin; CNU, cerebral nuclei; CTXsp, cortical subplate; DC, degree centrality; DMN, default mode network; fb, fiber tracts; fMRI, functional magnetic resonance imaging; HY, hypothalamus; KNN, k-nearest neighbors; MB, midbrain; OLF, olfactory area; PC, participation coefficient; ROI, region of interest; RSI, reciprocal social interaction; TBR1, T-Brain-1; TBS, theta-burst stimulation; TH, thalamus; WT, wild-type.

## Introduction

Autism spectrum disorders (ASDs), one of the most highly prevalent neurodevelopmental disorders, are characterized by 2 core behavioral deficits that appear in early childhood; one is impaired social interaction and communication, and the other is restricted, repetitive behavioral patterns and sensory abnormalities (https://dsm.psychiatryonline.org/doi/book/10.1176/appi.books.9780890425787). Seeking the biological mechanisms causing these core behavioral manifestations is crucial for providing a better understanding of ASD and uncovering potential treatments. One of the primary biological features correlated with behavioral phenotypes in ASD patients is the altered neural connections [1–4] contributing to both local and long-range functional connectivity [5–8]. Some mouse models of ASD also present connectivity deficits [6,9–14]. However, the detailed features of rewiring alterations under ASD conditions remain elusive.

T-Brain-1 (TBR1), a neuron-specific T-box transcription factor [15], has been identified as a causative gene of ASD (Gene score 1, https://gene.sfari.org/database/human-gene/TBR1) [16,17]. TBR1 regulates the expression of a set of genes controlling axonal projections of cortical and amygdalar neurons [17,18]. Contralateral projection of basolateral amygdalae (BLA) neurons via the posterior part of the anterior commissure is particularly sensitive to *Tbr1* deficiency [17], with this latter eliciting anterior commissure impairment in both humans and mice [17,19–21]. This evolutionarily conserved connection defect makes mice an excellent model for TBR1-linked autism. Indeed, *Tbr1* haploinsufficiency, a condition mimicking genetic variations observed in patients with ASD, results in many autism-like behavioral abnormalities in mice, including reduced social behaviors, defective vocalization, impaired olfactory discrimination, aversive memory, and cognitive inflexibility [17,18,22]. Disconnection of the anterior commissure in mice either surgically or via unilateral chemogenetic inhibition of the BLA results in autism-like behaviors [23,24]. Together, these studies have demonstrated that *Tbr1*$^{+/-}$ mice may serve as an excellent model for exploring the behavioral and connectivity deficits associated with ASD.

Given that BLA axonal projection is particularly sensitive to *Tbr1* haploinsufficiency [17,23] and that BLA activation by local infusion of D-cycloserine, a co-agonist of NMDAR, is sufficient to ameliorate the autism-like behaviors exhibited by *Tbr1*$^{+/-}$ mice [17], we speculate that BLA-derived circuits are critical to *TBR1*-linked ASD phenotypes. To explore that possibility, we established a whole-brain analytical platform using serial mouse brain sections for immunostaining and high-content imaging and for performing registration and quantification based on the Allen Mouse Common Coordinate Framework version 3 (CCFv3) [25]. We injected adeno-associated virus (AAV) expressing channelrhodopsin (ChR) fused with fluorescent protein into the BLA of mice and analyzed axonal projection of the BLA at the mesoscale level based on fluorescent protein signals and then assessed neuronal activity after blue light stimulation based on C-FOS expression at a cellular level. Our high-resolution analyses revealed the altered inter- and intrahemispheric connectopathy in *Tbr1*$^{+/-}$ mice, providing clear evidence that an ASD-linked condition results in dyssynchronization at the whole-brain scale. Moreover, optogenetic stimulation at BLA enhanced whole-brain synchronization and increased nose-to-nose investigations by *Tbr1*$^{+/-}$ mice of conspecific unfamiliar mice, implying that deep brain stimulation of amygdala may represent a potential treatment for TBR1-linked autism.

## Results

### Development of a mesoscopic whole-brain scale circuit analysis pipeline

To analyze structural and functional BLA connectivity in individual mice, we unilaterally injected AAV expressing the ultrafast ChR variant oChIEF fused with the fluorescent protein

Citrine at BLA of both *Tbr1*$^{+/-}$ and wild-type (WT) mice (**Fig 1A**). Given that oChIEF-Citrine targets the plasma membrane of entire neurons, in addition to being activated by blue light, oChIEF-Citrine was also used to outline BLA neurons, including axons. The fluorescent signal of oChIEF-Critine outside of the BLA reflects the extent of axonal projections derived from unilateral BLA neurons. To analyze the functional connectivity of the BLA, *Tbr1*$^{+/-}$ and WT mice were further separated into 2 groups. One group unilaterally received optical theta-burst stimulation (TBS) at the BLA of the AAV-injected side when they were freely moving in their home cages. The other control group was subjected to the same experimental procedure except for optical stimulation. Then, we analyzed neuronal activation by BLA stimulation based on C-FOS expression 2 h after stimulation (**Fig 1A and 1B**).

The fluorescence images of the serial whole-brain sections were then subjected to quantification at a whole-brain scale (**Fig 1C–1E**). Our entire pipeline basically comprises 5 elements, which have been outlined and detailed in **Figs 1 and S1** and the **Methods**. The list of brain regions used in our study and their corresponding abbreviations can be found in **S1 Data**. Brain ontology is identical to CCFv3.

## Structure-wise comparisons reveal rewiring in *Tbr1*$^{+/-}$ mouse brains

First, we analyzed the whole-brain distribution pattern of oChIEF-Citrine. We found that oChIEF-Citrine signals were widely distributed ipsilaterally in WT brains and also extended contralaterally (**Fig 2A** and **S1 Video**, upper). However, oChIEF-Citrine signals barely extended contralaterally in *Tbr1*$^{+/-}$ mouse brains (**Fig 2A** and **S1 Video**, lower).

We compared brains from a total of 14 *Tbr1*$^{+/-}$ mice and 13 WT littermates. Since the BLA consists of heterologous projections with varying patterns of long-range axonal projection [26], first we investigated the distributions of the virus injection sites between WT and *Tbr1*$^{+/-}$ mouse brains and confirmed that they were indistinguishable and comparable (**S2A–S2D Fig**). Next, we binned and summarized the average percentage of oChIEF-Citrine signals in all brain regions of *Tbr1*$^{+/-}$ mice and WT littermates. These results were subjected to 3D projection at both coronal and horizontal views, which indicated a reduction in contralateral projections of BLA neurons in *Tbr1*$^{+/-}$ mice (**Fig 2B** and **S2 Video**).

To quantify the difference in BLA projections, we performed a slice-based analysis to determine the distribution of Citrine signal along the anterior-posterior (AP) axis of every brain region in each slice. A structure-wise comparison between *Tbr1*$^{+/-}$ mice and WT littermates was then performed at the whole-brain scale. The Citrine signals in the ipsilateral, i.e., the virus injection site, and contralateral hemispheres were quantified separately (**S3 Fig** left, **S1 Data**). Consistent with our previous findings [17,23], we found that oChIEF-Citrine signals were greatly reduced in contralateral BLA-related regions (i.e., LA, BLAa, BLAp), and in the associated fiber tracts (act, amc), of *Tbr1*$^{+/-}$ mice compared to those of WT littermates (**Fig 2C–2E and S1 Data**), supporting the reliability of our whole-brain analysis system.

Using our whole-brain structure-wise quantification, we found that a total of 104 brain regions exhibited differential oChIEF-Citrine expression at the contralateral site (**Fig 2C–2E**). Among them, 56 regions had much lower oChIEF-Citrine signals in *Tbr1*$^{+/-}$ mice relative to WT, and these regions mainly reside in areas of the forebrain, such as the isocortex, olfactory area (OLF), cortical subplate (CTXsp), cerebral nuclei (CNU), and fiber tracts (fb) (**Fig 2C–2E**). We term these regions "affected contralateral projection regions (ACPRs)". In WT mice, these regions receive topographically bilateral projections from the 2 amygdalae in the 2 brain hemispheres (**Fig 2A and 2B,** upper panel; also see **Allen Mouse Brain Connectivity Atlas**: connectivity.brain-map-org/projection/experiment/113144533 and connectivity.brain-map. org/projection/experiment/277710753). The other 48 regions had much higher

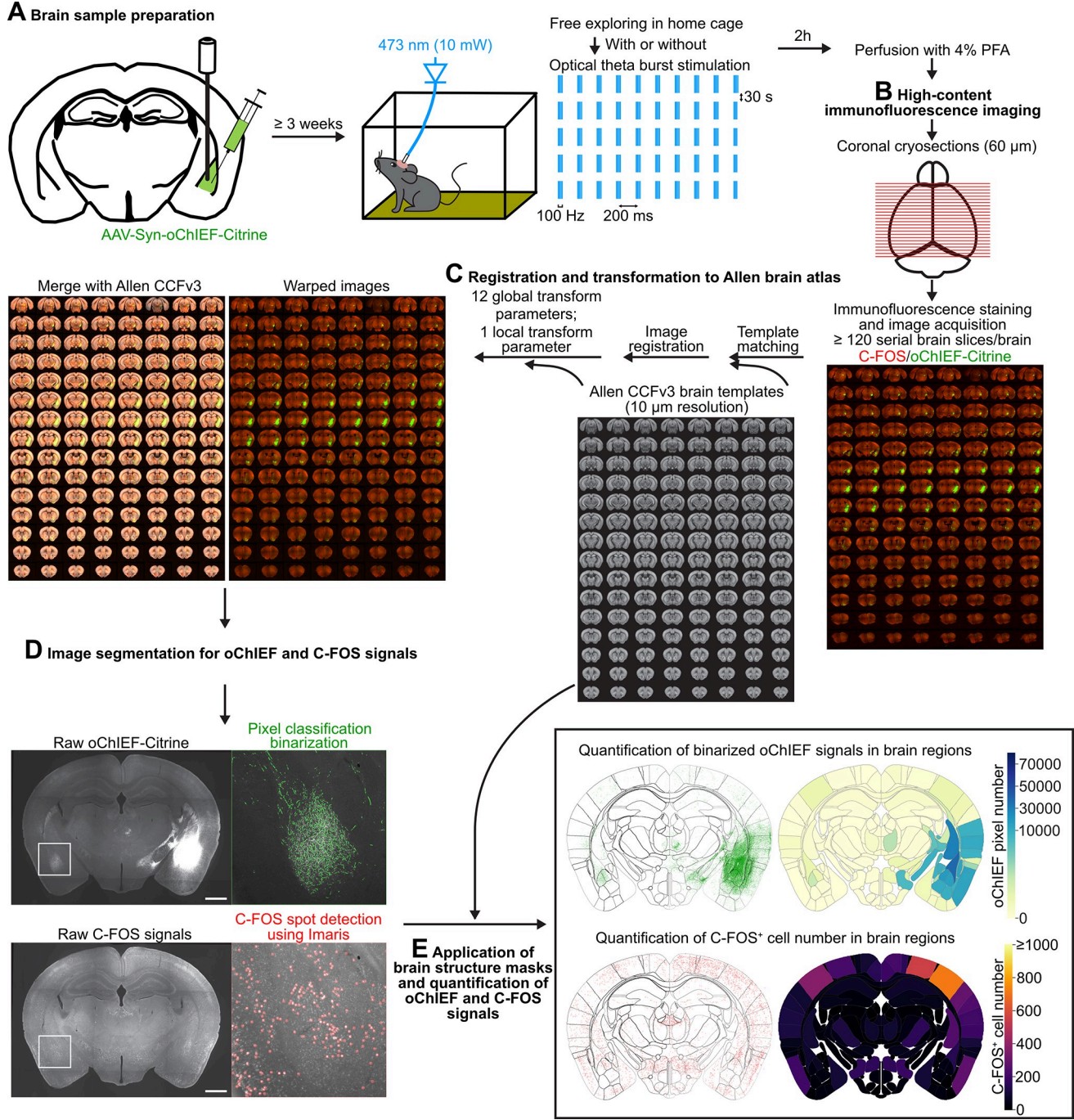

**Fig 1. Workflow of whole-brain-scale analysis of BLA-derived circuits.** There are 5 steps in the analysis pipeline. (**A**) Brain sample preparation, involving unilateral BLA infection of oChIEF-Citrine-expressing AAV, in vivo TBS of the AAV-infected BLA while the awake mice are in their home cage, and brain fixation by PFA 2 h after optogenetic stimulation. (**B**) High-content immunofluorescence imaging. At least 120 coronal cryosections (60 μm/section) of a single mouse brain were subjected to immunostaining to reveal C-FOS+ cells and oChIEF-Citrine signals and then imaged by a high-content imaging system. (**C**) Registration and transformation to the Allen brain atlas. Raw image series were registered to the templates of Allen CCFv3 by sequentially applying 12 global and 1 local transform parameters. (**D**) Image segmentation for oChIEF and C-FOS signals. The oChIEF signals were isolated by pixel classification and binarized. C-FOS+ cell locations were detected by the spot detection function of Imaris (also see **S1 Fig**). (**E**) Application of brain structure masks and quantification of oChIEF and C-FOS signals. Examples of choropleth maps show the quantification of oChIEF+ pixel number and C-FOS+ cell number of different brain regions in a single coronal section. Scale bar in (**D**), 1 mm. AAV, adeno-associated virus; BLA, basolateral amygdala; CCFv3, Common Coordinate Framework version 3; PFA, paraformaldehyde; TBS, theta-burst stimulation.

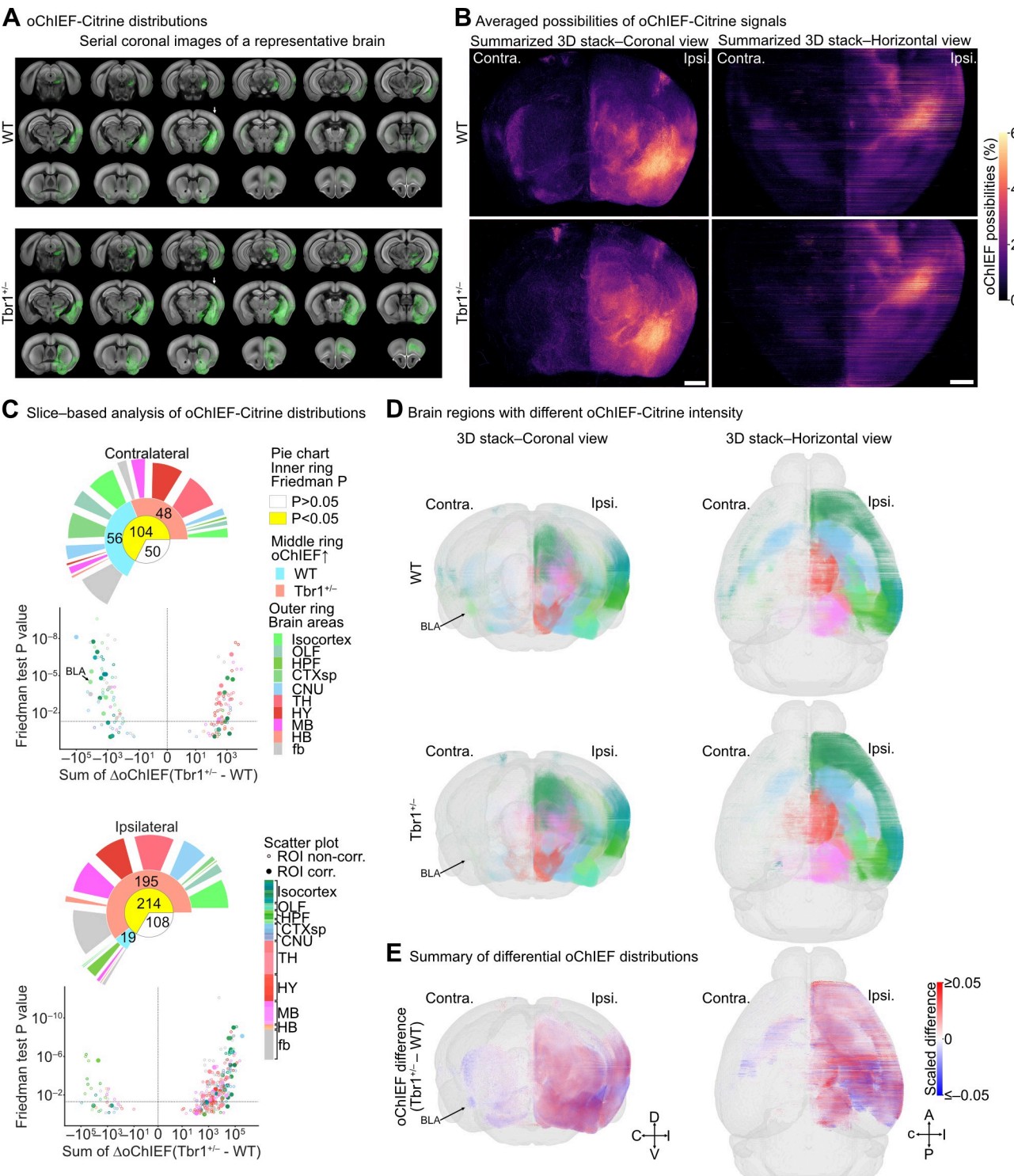

**Fig 2. Rewiring of BLA-derived circuits.** (**A**) Coronal image series of binarized oChIEF-Citrine–labeled axonal signals (green) superimposed on a CCFv3 template (gray) of a WT (upper) and a *Tbr1*^+/− mouse brain (lower) (also see **S1 Video**). (**B**) Coronal (left) and horizontal (right) views of summarized 3D stacks with binarized and binned (10 μm³) oChIEF signals from WT (upper) and *Tbr1*^+/− (lower) mouse brains. Color map indicates different probabilities of oChIEF signal in a 10-μm³ cube (also see **S2 Video**). (**C**) Structure-wise comparison of binarized oChIEF signal distributions along the AP axis between WT and *Tbr1*^+/− mouse brains after slice-based quantification (also see **S1 Data**). Scatterplots of slice-based analysis show the summation of oChIEF differences along the AP axis (sum of ΔoChIEF) versus significance (Friedman test *P* value, determined by Friedman one-way repeated measure analysis) for each brain region. Vertical and horizontal dotted lines represent the sum of ΔoChIEF = 0 and Friedman test *P*

value = 0.05, respectively. Open and filled circles represent ROI noncorrected and corrected brain regions, respectively. Colors represent brain region identities. The arrow points to the filled circle representing the contralateral BLA. (**D**) Coronal (left) and horizontal (right) views of 3D maps with binarized and binned (10 μm$^3$) oChIEF pixels in the brain regions displaying significantly different BLA-circuit rewiring ($P < 0.05$) between WT (upper) and *Tbr1*$^{+/-}$ mice (lower). Colors of oChIEF pixels in (**D**) represent brain regions as shown in (**C**) and the alpha levels (0–1) of the brain region color represent scaled oChIEF probabilities. Arrows point to the binned oChIEF pixels within the contralateral BLA. (**E**) Coronal (left) and horizontal (right) views of 3D maps showing the difference of binarized and binned (10 μm$^3$) oChIEF pixels (Δ oChIEF) between WT and *Tbr1*$^{+/-}$ mouse brains. Heatmap of the scaled difference indicates the differential oChIEF signal probabilities (ΔoChIEF of *Tbr1*$^{+/-}$–WT). Arrow highlights that Tbr1$^{+/-}$mice display massive loss of contralateral axonal projections in the BLA. Scale bar in (**B**), 1 mm. Sample sizes of mice are the same for (**B–E**) (WT, $n = 13$; *Tbr1*$^{+/-}$, $n = 14$). The numerical value data for plotting (**C**) are available in **S1 Data**. AP, anterior-posterior; BLA, basolateral amygdala; CCFv3, Common Coordinate Framework version 3; ROI, region of interest; WT, wild-type.

oChIEF-Citrine signals in *Tbr1*$^{+/-}$ mice compared to WT mice (**Fig 2C–2E**). These regions were mainly localized in the thalamus (TH), hypothalamus (HY), and midbrain (MB), which were mistargeted by contralateral BLA projection in *Tbr1*$^{+/-}$ mice (**Fig 2C–2E**).

At the ipsilateral site, we uncovered twice as many brain regions displaying different oChIEF-Citrine levels. We found that 195 brain regions had increased oChIEF-Citrine signals in *Tbr1*$^{+/-}$ mouse brains (**Fig 2C–2E**), which we denote "affected ipsilateral projection regions (AIPRs)". Note that the majority of AIPRs are not the counterparts of the ACPRs, such as MOs, ORBl, ORBvl, ACAv, FRP, ILA, and PPN (ΔoChIEF (Tbr1$^{+/-}$–WT) > 10,000) (**Fig 2C–2E**). We even found that 19 ipsilateral regions had decreased oChIEF-Citrine signals in *Tbr1*$^{+/-}$ mouse brains (**Fig 2C–2E**). These findings indicate that *Tbr1* haploinsufficiency impairs both contra-lateral and ipsilateral projections of BLA neurons and leads to mistargeting to ipsilateral brain regions and not necessarily limited to ACPR counterparts. We also noticed that AIPRs widely encompass several regions in the isocortex, TH, HY, MB, and fb (**Fig 2C–2E**), implying that *Tbr1* haploinsufficiency may elicit stronger BLA ipsilateral connections with those regions and influence neuronal activity at the ipsilateral side.

## Altered functional outcomes of rewired BLA-derived circuits in *Tbr1*$^{+/-}$ mouse brains

In addition to structural connections, we further analyzed the whole-brain modulation of neu-ronal activities upon unilateral BLA activation, representing the net effect of direct or indirect BLA innervation. Modulation of neuronal activity was based on C-FOS$^+$ cell numbers in the absence or presence of TBS treatment at BLA. We adapted our slice-based quantification to determine C-FOS$^+$ cell numbers along the AP axis (with 100 μm binning) of both brain hemi-spheres (**S3 Fig**, right). Pairwise comparisons of our 4 experimental groups i.e., WT ctrl, *Tbr1*$^{+/-}$ ctrl, WT TBS, and *Tbr1*$^{+/-}$ TBS, were then conducted to establish which brain regions exhibited differential C-FOS expression (**Fig 3** and **S1 Data**). These comparisons allowed us to analyze functional connections derived from the BLA at a whole-brain mesoscopic scale.

First, we compared the *Tbr1*$^{+/-}$ ctrl versus WT ctrl groups. Although *Tbr1*$^{+/-}$ mice had fewer BLA contralateral projections to many forebrain regions, we unexpectedly uncovered that only 5 regions displayed fewer C-FOS$^+$ cells in the *Tbr1*$^{+/-}$ ctrl group relative to WT ctrl. In contrast, 104 regions had more C-FOS$^+$ cells in the contralateral side of *Tbr1*$^{+/-}$ ctrl group brains (**Fig 3A**). On the ipsilateral side, we identified 94 regions having more C-FOS$^+$ cells and only 5 regions having fewer C-FOS$^+$ cells in the *Tbr1*$^{+/-}$ ctrl group compared to the WT ctrl group (**Fig 3A**). Thus, generally speaking, the *Tbr1*$^{+/-}$ ctrl group exhibited increased C-FOS$^+$ cell numbers compared to the WT ctrl group (**Fig 3B**), although the mechanism causing this increased cell number is unclear. Nevertheless, these differentially activated regions are widely distributed across the brain, from the isocortex to MB, suggesting that the abnormally high brain activities of *Tbr1*$^{+/-}$ mice during wakefulness (the state of our mice under experimental/

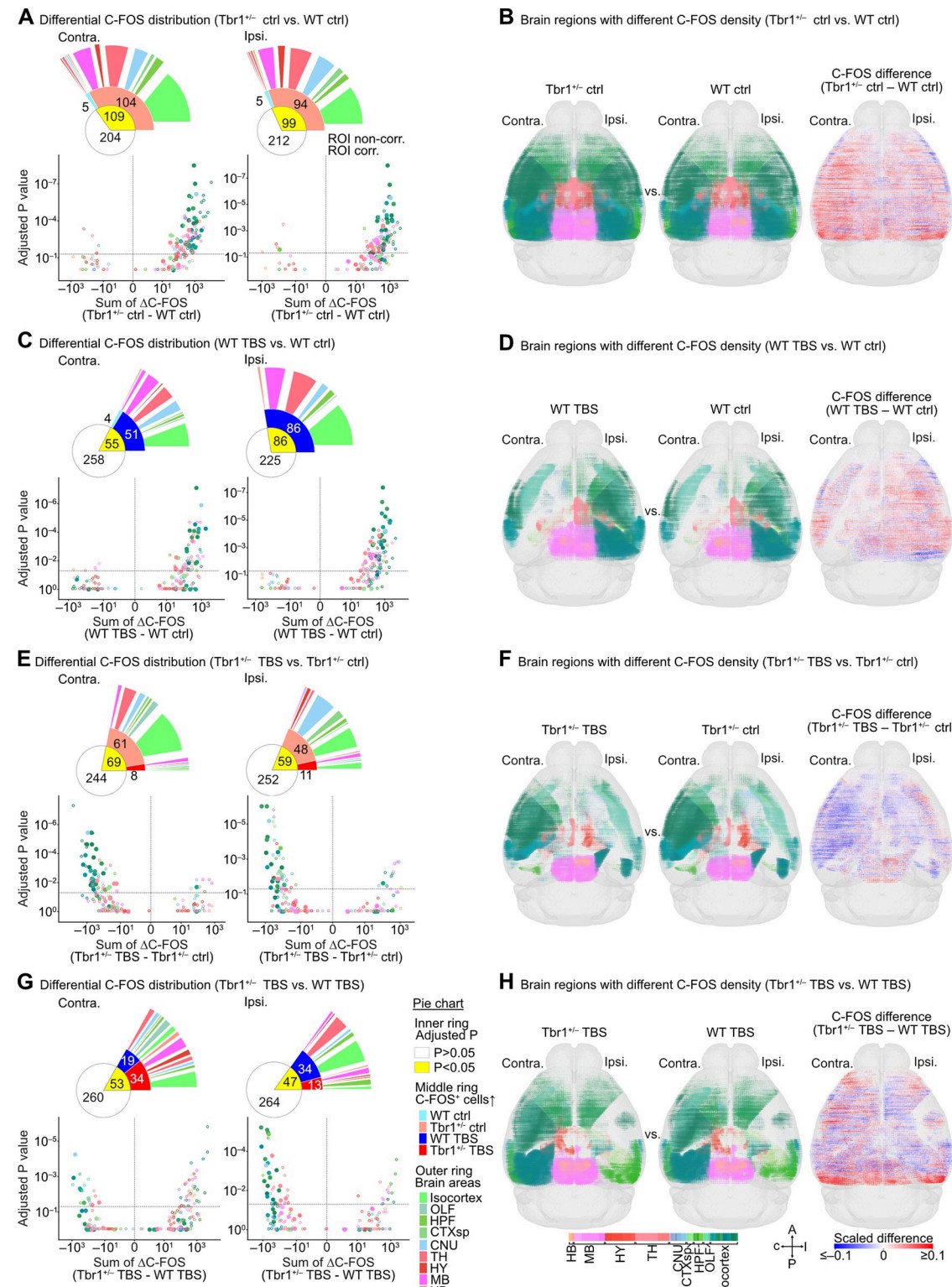

**Fig 3. Differential C-FOS expression in WT and *Tbr1*^+/− mouse brains with or without activation of BLA-derived circuits. (A, C, E, G)** Structure-wise comparison of C-FOS^+ cell number distributions along the AP axis in the contralateral (left) and ipsilateral (right) brain hemispheres of (**A**) *Tbr1*^+/− ctrl vs. WT ctrl, (**C**) WT TBS vs. WT ctrl, (**E**) *Tbr1*^+/− TBS vs. *Tbr1*^+/− ctrl, and (**G**) *Tbr1*^+/− TBS vs. WT TBS groups. Each scatterplot shows the summed difference in C-FOS^+ cell number long the AP axis (sum of ΔC-FOS) versus statistical significance (adjusted *P* value) for each brain region of the pairwise-compared groups (also see **S1 Data**). Friedman one-way repeated measure analyses, followed by post hoc pairwise tests with Bonferroni correction, were used for statistical analysis.

Only brain regions with a significant difference as determined by Friedman one-way repeated measure analyses are shown. Vertical and horizontal dotted lines represent the sum of ΔC-FOS = 0 and adjusted *P* value = 0.05, respectively. Colors represent brain region identities (indicated in **G**). Open and filled circles represent ROI-noncorrected and ROI-corrected brain regions, respectively. The pie charts illustrate the proportions of brain regions that fit the condition annotated in the right panel of (**G**). (**B, D, F, H**) Horizontal views of 3D maps with binned (100 μm$^3$) C-FOS cubes in the brain regions displaying a significant difference (adjusted *P* < 0.05) between compared groups. The left and middle panels are the results of individual conditions, as indicated. C-FOS$^+$ cubes are shown in brain region color codes (indicated in **H**), with their sizes representing the relative normalized density of C-FOS$^+$ cells within binned cubes. The right panels show the differences in the density of C-FOS$^+$ cells (ΔC-FOS, 100 μm$^3$ bins). The scale of the heatmap for ΔC-FOS is shown in (**H**). Sample sizes of mice are identical in (**A–H**) (WT ctrl, *n* = 6; *Tbr1*$^{+/-}$ ctrl, *n* = 7; WT TBS, *n* = 7; *Tbr1*$^{+/-}$ TBS, *n* = 7). The numerical value data for plotting (**A, C, E, G**) are available in **S1 Data**. AP, anterior-posterior; BLA, basolateral amygdala; ROI, region of interest; TBS, theta-burst stimulation; WT, wild-type.

control treatment) can be linked to circuits and areas distributed across the entire brain (**Fig 3B**).

Next, we analyzed the impact of BLA activation on whole-brain modulation of C-FOS expression. TBS at the BLA increased the number of C-FOS$^+$ cells in 51 contralateral regions and 86 ipsilateral regions of WT mice. Only 4 regions in the contralateral side presented reduced C-FOS$^+$ cell numbers in the WT TBS group (**Fig 3C**, WT TBS versus WT ctrl). Thus, unilateral activation of the BLA in WT mice mainly activates other brain regions, both contralaterally and ipsilaterally (**Fig 3D**). In contrast, *Tbr1*$^{+/-}$ mice displayed fewer C-FOS$^+$ cells at 61 contralateral and 48 ipsilateral regions upon TBS at BLA. Only 8 and 11 regions in the contralateral and ipsilateral sides, respectively, had increased C-FOS$^+$ cell numbers in *Tbr1*$^{+/-}$ mice (**Fig 3E**, *Tbr1*$^{+/-}$ TBS versus *Tbr1*$^{+/-}$ ctrl). These results suggest that *Tbr1* deficiency generally represses neuronal activation of other brain regions upon TBS at BLA (**Fig 3F**), and the opposite is true for WT mice.

We also detected differing C-FOS$^+$ cell distributions in many brain regions between the *Tbr1*$^{+/-}$ TBS and WT TBS groups. The numbers of differentially activated/inhibited regions were fewer than for the other 3 comparisons described above. Moreover, we uncovered a reduced propensity for either increased or decreased C-FOS$^+$ cell numbers in WT and Tbr1$^{+/-}$ mice upon TBS at BLA (**Fig 3G**, *Tbr1*$^{+/-}$ TBS versus WT TBS). We identified 19 regions with reduced and 34 with enhanced C-FOS$^+$ cell numbers at the contralateral side of *Tbr1*$^{+/-}$ mice. On the ipsilateral side, 34 and 13 regions presented reduced and increased C-FOS$^+$ cell numbers, respectively (**Fig 3H**). Thus, this comparison indicates that TBS at BLA of *Tbr1*$^{+/-}$ mice elicits a different outcome from that of WT mice.

Taken together, these results imply that *Tbr1* deficiency alters functional connections between BLA and other brain regions and it plays a critical role in whole-brain responses to stimulation at the BLA.

## Correlations between rewired BLA projection and altered C-FOS expression patterns

Several possibilities may explain the observed differences between structural and functional connections of WT and *Tbr1*$^{+/-}$ mice described above. One is an indirect influence of targets directly innervated by BLA. Another is a dominant influence of local microcircuits (such as feed-forward or feedback inhibitions), regardless of receiving BLA axonal targeting. Significantly, we found that although many regions in the contralateral TH of *Tbr1*$^{+/-}$ mice had more BLA axonal projections, the numbers of C-FOS$^+$ cells were not increased but somehow decreased after unilateral TBS at BLA (**Figs 2C** and **3E**). To dissect further the correlation between BLA axonal projection and neural activation of specific brain regions, we selected the brain regions exhibiting rewired axonal innervations and altered C-FOS expression patterns upon TBS of the BLA in either WT or *Tbr1*$^{+/-}$ mice for further analysis.

First, we separately determined contralateral and ipsilateral ΔoChIEF traces by pairwise subtraction of oChIEF-Citrine signals along the AP axis (100 μm binning) between *Tbr1*$^{+/-}$ and WT samples (**Fig 4A and 4C,** left, subtraction of coexisting 10 μm$^3$ binning cubes between *Tbr1*$^{+/-}$ and WT samples for visualization; **Fig 4B** and **4D**, middle row, ΔoChIEF traces). Neural activation induced by TBS in *Tbr1*$^{+/-}$ and WT mice, i.e., *Tbr1*$^{+/-}$ ΔC-FOS traces and WT ΔC-FOS traces, respectively, was also determined. Pairwise subtraction between *Tbr1*$^{+/-}$ ΔC-FOS and WT ΔC-FOS traces was then executed to obtain all possible differences, i.e., ΔΔC-FOS traces (**Fig 4A** and **4C**, middle, subtractions of coexisting 100 μm$^3$ binning cubes for visualization; **Fig 4B** and **4D**, middle row, ΔΔC-FOS traces). A Pearson correlation coefficient (r) was calculated according to the correlation of the averaged ΔoChIEF trace and the averaged ΔΔC-FOS trace in each brain region. The brain regions exhibiting positive correlations between rewired BLA-derived circuits and altered C-FOS expression patterns reflect concurrent changes to BLA-derived axonal innervations and their neural activity upon BLA activation. In the contralateral hemisphere of *Tbr1*$^{+/-}$ mice, the AUD regions (AUD, AUDp), AIp, PIR, BLA, and MBsen regions all exhibited impaired BLA contralateral axonal projections and consistently resulted in reduced boosting of BLA activation on C-FOS expression, suggesting that reduced BLA axonal projection likely contributes to a loss of intensification effect of contralateral BLA input to these regions in the *Tbr1*$^{+/-}$ mice (**Fig 4A** and **4B**, positive correlation). In contrast, the contralateral thalamus, DORsm regions (DORsm, VENT), and ILM each displayed higher oChIEF-Citrine signals in *Tbr1*$^{+/-}$ mice. However, more BLA axonal innervation did not result in more C-FOS$^+$ cells in these regions. Thus, they exhibited negative correlations between ΔoChIEF and ΔΔC-FOS (**Fig 4A** and **4B**). Local inhibitory microcircuits triggered by mistargeting BLA inputs in these regions may mediate the net inhibitory effect. Alternatively, indirect innervation by direct BLA targets of other brain regions may also contribute to the inhibitory effect.

In the ipsilateral hemisphere, the correlation results between ΔoChIEF and ΔΔC-FOS were the opposite of those for the contralateral hemisphere (**Fig 4A** and **4B** versus **Fig 4C** and **4D**). Only the posterior part (ventral) of the HIP and NOT had reduced ipsilateral BLA axonal projections in *Tbr1*$^{+/-}$ mice, and it exhibited impaired boosting of BLA activation on C-FOS expression (**Fig 4C** and **4D**, left panel). The reduction in direct innervation of the BLA likely contributed to the decreased neuronal activation of these 2 regions. A major group of regions at the ipsilateral side exhibited a higher level of oChIEF-Citrine in *Tbr1*$^{+/-}$ mice but a negative correlation between ΔoChIEF and ΔΔC-FOS (**Fig 4C** and **4D**). This group includes the isocortical area (MOp, AUDd, ACA, ACAv, TEa), as well as the CNU (MEA, PALv, SI, PALc, BST), TH (DORsm, DORpm, CL), HY (LZ), and MB (SCdg, VTA) areas. Again, local microcircuits are likely involved in the functional alterations among BLA and these regions in *Tbr1*$^{+/-}$ mice. Mistargeting of BLA axons in *Tbr1*$^{+/-}$ mice may lead to incorrect innervation and initiates abnormal local microcircuits to alter functional outcomes.

Overall, the results of this correlation analysis reveal that the structural rewiring of BLA-derived circuits caused by *Tbr1* deficiency triggers complex alterations in the functional outcomes of neuronal activities at a whole-brain scale.

## Alteration of whole-brain synchronization in *Tbr1*$^{+/-}$ mice

In addition to the paired comparison of functional alterations to individual structures described above, we also assessed changes in neuronal activity at the network level. We evaluated functional connectivity, which is defined by the synchronization of C-FOS expression levels between 2 brain regions across all subjects [27,28]. To do so, we first determined the C-FOS$^+$ cell density of individual brain regions (**S4 Fig**, volume-based quantification). The

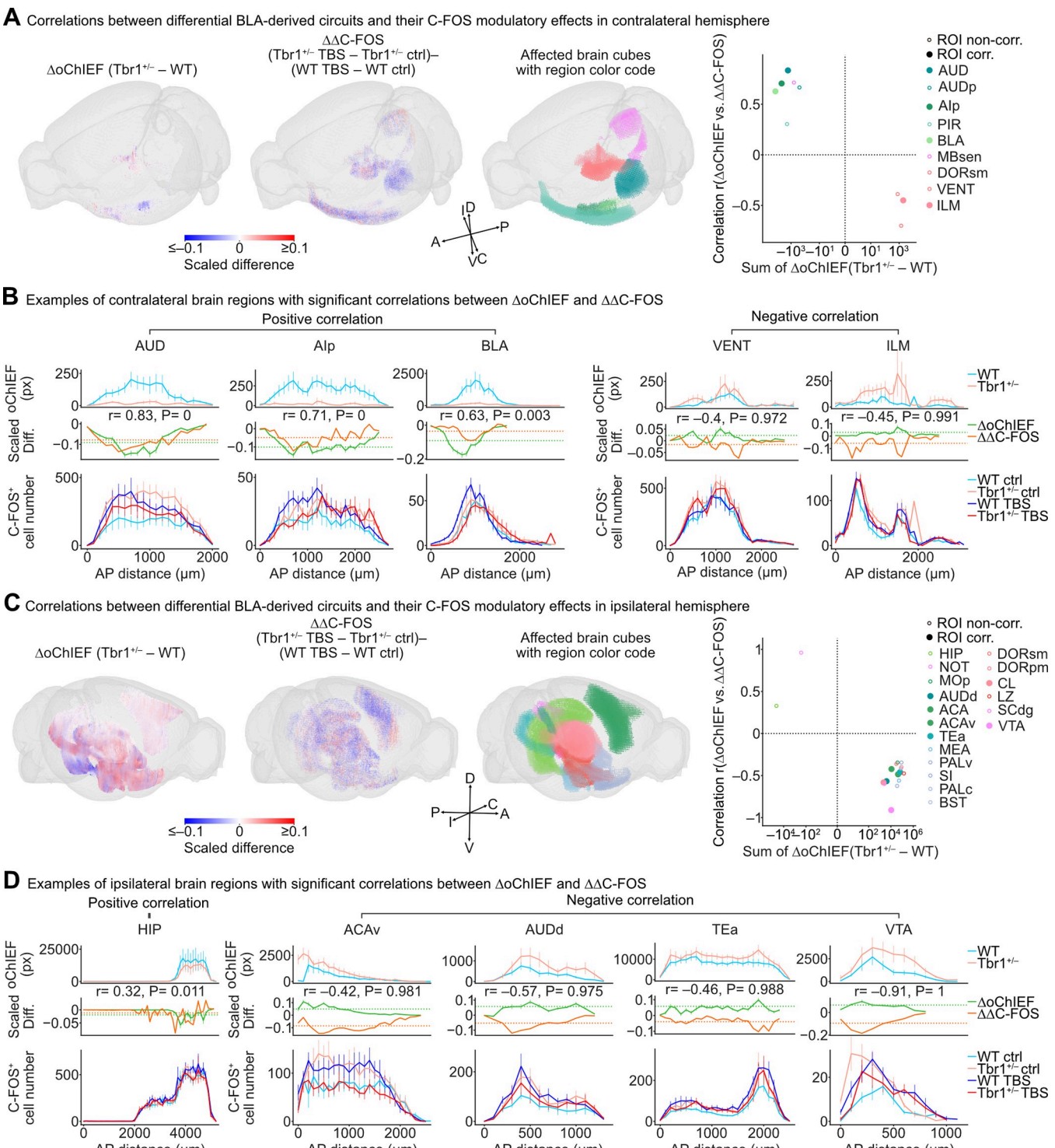

**Fig 4. Correlations of BLA axonal projection patterns and C-FOS expression.** (**A**) The 3D maps of rewired BLA-derived circuits (left, 10 μm³ binned cubes with binarized oChIEF signals) and differentially TBS-modulated C-FOS expression levels (middle, 100 μm³ binned C-FOS⁺ cell numbers) in affected brain regions (right) with the Allen brain region color code. Only the brain regions showing significant correlations between rewired BLA-derived circuits (indicated by ΔoChIEF) and altered C-FOS expression patterns upon TBS of BLA-derived circuits (indicated by ΔΔC-FOS) of the contralateral hemisphere are shown here. Heatmap of the scaled difference shows the differential oChIEF signal probabilities (left 3D map, ΔoChIEF of *Tbr1⁺/⁻*–WT) and the difference between the C-FOS modulatory effects of BLA-derived circuits in *Tbr1⁺/⁻* (ΔC-FOS of *Tbr1⁺/⁻* TBS–*Tbr1⁺/⁻* ctrl) and WT (ΔC-FOS of WT TBS–WT ctrl) mouse brains (middle left 3D map, ΔΔC-FOS). Scatterplots in the right panels show the sum of ΔoChIEF (*Tbr1⁺/⁻*–WT) versus the correlation between ΔoChIEF and

ΔΔC-FOS for the brain regions of the contralateral hemispheres. Colors indicate brain region identities. Open and filled circles represent ROI-noncorrected and ROI-corrected brain regions, respectively. (**B**) Examples of brain regions that have positive or negative correlations between rewired BLA-derived circuits of *Tbr1*⁺/⁻ mice and differential TBS-derived C-FOS expression levels. The oChIEF distributions (WT and *Tbr1*⁺/⁻, upper panel), C-FOS⁺ cell distributions (WT ctrl, *Tbr1*⁺/⁻ ctrl, WT TBS, and *Tbr1*⁺/⁻ TBS, lower panel), and scaled differences (ΔoChIEF and ΔΔC-FOS, middle panel) along the AP axis are shown. Dashed lines in plots of scaled differences represent the average 99th percentile of scaled differences calculated from the pair-subtracted population data with 1,000 times AP axis reshuffling. (**C**) The 3D maps of the results of the ipsilateral hemisphere. (**D**) Examples of brain regions that exhibit correlations between ΔoChIEF and ΔΔC-FOS signals in the ipsilateral hemisphere. Data represent mean ± SEM. Contralateral (**A**, **B**) and ipsilateral (**C**, **D**) hemispheres were analyzed separately. Sample sizes are the same in (**A**–**D**) (oChIEF data: WT, *n* = 13; *Tbr1*⁺/⁻, *n* = 14. C-FOS data: WT ctrl, *n* = 6; *Tbr1*⁺/⁻ ctrl, *n* = 7; WT TBS, *n* = 7; *Tbr1*⁺/⁻ TBS, *n* = 7). The numerical value data are available in **S2 Data**. AP, anterior-posterior; BLA, basolateral amygdala; ROI, region of interest; TBS, theta-burst stimulation; WT, wild-type.

interregional correlation coefficient of C-FOS levels, r, was then calculated to reflect synchronization level and considered meaningful if its significant correlation level *P* was <0.05; otherwise, r was set to 0 to reflect no correlation. Accordingly, we identified large proportions of interregional pairs with positive correlations and comparable positive r values in all 4 experimental groups of WT ctrl, *Tbr1*⁺/⁻ ctrl, WT TBS, and *Tbr1*⁺/⁻ TBS, yet very few but variable interregional pairs exhibiting negative correlations in the WT ctrl (1.52%), *Tbr1*⁺/⁻ ctrl (2%), WT TBS (2.36%), and *Tbr1*⁺/⁻ TBS (0.03%) groups (**Fig 5A**). An interregional correlation matrix of C-FOS levels was then computed for each experimental group (**Fig 5B**).

Upon examining the C-FOS correlation matrix, we observed that meaningful interregional correlations between brain regions in *Tbr1*⁺/⁻ mice in the awake state were noticeably reduced compared to WT mice (**Fig 5B**, WT ctrl versus *Tbr1*⁺/⁻ ctrl). After TBS, interregional correlations were noticeably reduced in the WT group (**Fig 5B**, WT ctrl versus WT TBS), whereas they were dramatically increased in *Tbr1*⁺/⁻ mice (**Fig 5B**, *Tbr1*⁺/⁻ ctrl versus *Tbr1*⁺/⁻ TBS). Therefore, the opposing consequences caused by BLA activation in WT and *Tbr1*⁺/⁻ mice are not only reflected in the neuronal recruitment of individual structures but also manifested by interregional activity correlation.

We further used network analysis for each C-FOS correlation matrix and quantified the degree centrality (DC) of each brain region based on the data of interregional C-FOS correlations to evaluate the differences among our 4 experimental groups (**S5A Fig**). A higher DC for a specific region indicates that the activity of that region is synchronous with more brain regions. Our results show that the WT ctrl group had a higher DC than the *Tbr1*⁺/⁻ ctrl group. However, TBS at the BLA reduced DC across the entire brain in WT mice but increased it in *Tbr1*⁺/⁻ mice (**S5A Fig**). Thus, the DC analysis also suggests that the WT ctrl group was more synchronous than the *Tbr1*⁺/⁻ ctrl group and that TBS at the BLA results in an opposing outcome in WT and *Tbr1*⁺/⁻ mice.

To confirm that the differences in DC were not caused by the particular correlation criteria we used, we evaluated the network stability of the 4 experimental groups by increasing the r (correlation coefficient) or *P* (correlation significance) threshold used for constructing the interregional correlation matrix (**Fig 5B**) and applying network analysis. Compared to the WT ctrl network, the reduced mean DC of the *Tbr1*⁺/⁻ctrl network still existed across all the absolute r and a large range of *P* threshold conditions. Moreover, the contrasting modulatory effects of BLA-derived circuit activation on DC between the WT and *Tbr1*⁺/⁻ networks also existed when all the absolute r and large range of *P* value thresholds were used for network construction (**S5B Fig**). These results support that the differences in DC among the 4 experimental groups are unbiased.

To visualize the synchronization among major brain areas in a 3D map, we further calculated the mean values of the interregional C-FOS correlations within and between the major brain areas. The major brain areas we examined are listed in **Fig 5C** [29]. In the 3D maps of mouse brains, it is clear that the correlation networks between and within areas of the major

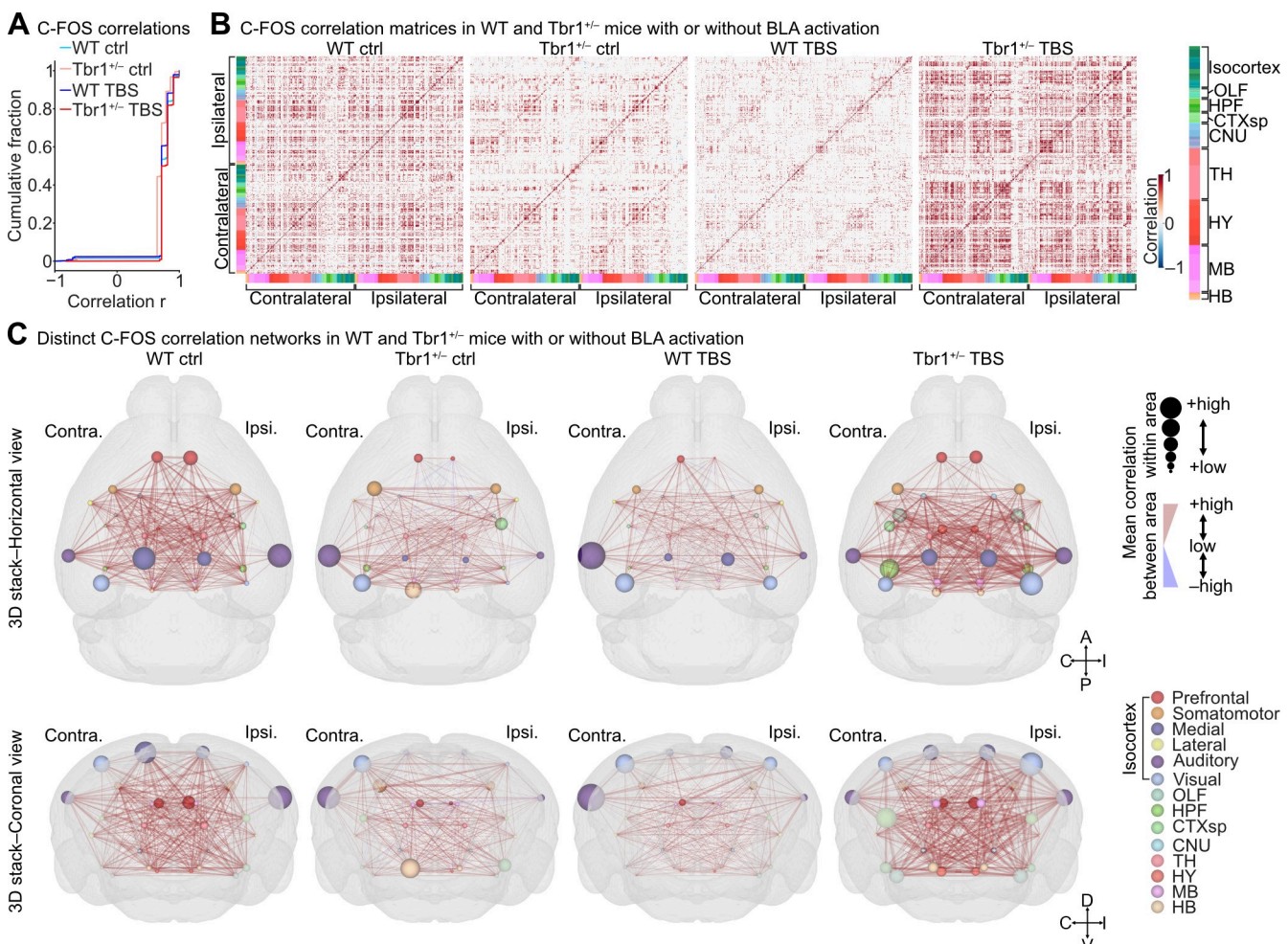

**Fig 5. Distinct interregional correlations of C-FOS expression in WT and *Tbr1*⁺/⁻ mouse brains with or without BLA-derived activation.** (**A**) Cumulative plot showing the probability of all significant interregional correlations ($P < 0.05$, Kendall's rank correlation) of C-FOS expression levels for the 4 experimental groups. (**B**) Matrices showing whole-brain interregional correlations of C-FOS expression in the WT ctrl, *Tbr1*⁺/⁻ ctrl, WT TBS, and *Tbr1*⁺/⁻ TBS groups (left to right). Brain regions are annotated with color codes (right panel). (**C**) Horizontal (top) and coronal (bottom) views of 3D maps showing simplified C-FOS correlation networks in the WT ctrl, *Tbr1*⁺/⁻ ctrl, WT TBS, and *Tbr1*⁺/⁻ TBS groups (left to right). Brain regions belonging to the same primary brain areas (colored spots, annotated in the lower right panel) have been combined. Mean correlations within and between primary brain areas were calculated and are expressed as spot size and edge thickness/color (annotated in the upper right panel), respectively. The sample sizes of mice are the same in (A–C) (WT ctrl, $n = 7$; *Tbr1*⁺/⁻ ctrl, $n = 8$, WT TBS, $n = 7$; *Tbr1*⁺/⁻ TBS, $n = 7$). The numerical value data are available in **S3 Data**. BLA, basolateral amygdala; TBS, theta-burst stimulation; WT, wild-type.

brain regions, especially the isocortex areas, were much weaker in the *Tbr1*⁺/⁻ ctrl group compared to the WT ctrl group (**Fig 5C**, left 2 panels), even though C-FOS cell density was higher in *Tbr1*⁺/⁻ mice (**Fig 3A**). This outcome also implies dyssynchronization of whole brain activity attributable to *Tbr1* deficiency. In contrast to the condition without TBS, unilateral TBS at the BLA of *Tbr1*⁺/⁻ mice generally reduced C-FOS⁺ cell density, but it resulted in better whole-brain synchronization compared to the *Tbr1*⁺/⁻ ctrl as well as WT TBS groups (**Fig 5C**). A majority of the brain areas of the *Tbr1*⁺/⁻ TBS group also exhibited stronger interregional correlations within areas (**Fig 5C**). Thus, our whole-brain C-FOS correlation analyses support that *Tbr1* deficiency impairs whole-brain synchronization in mice and that TBS at the BLA ameliorates, at least partially, the dyssynchronization caused by *Tbr1* deficiency.

## Differential modular organization and reorganization upon BLA activation

To further understand which brain regions are more densely interconnected than others and achieve better modular integration, we used a Louvain community detection algorithm [30] to analyze modular organization. Optimal communities were defined by C-FOS levels of the brain regions being more correlated to each other within a community than between communities. To determine a spatial resolution parameter, R, for each network, we used the range from 0.4 to 0.9 in Louvain community detection and calculated the difference in the modularity metric (Q) for communities representing real networks or shuffled networks. The appropriate R-value for each network was determined according to the maximal difference between Q and $Q_{shuffled}$ (**S6A Fig**) and used for Louvain community detection.

We found that the modular organizations of our experimental groups were distinct. The WT ctrl network contained 14 communities, with the majority of brain regions occupying the first 3 communities (598/618 brain regions). The minor communities of the WT ctrl network comprised a few intercorrelated regions (communities 4, 6, 8, 9) or single brain regions (communities 5, 7, 10, 11, 12, 13, 14) (**Figs 6A, 6B, S6B, and S7 and S3 Data**). In contrast, the $Tbr1^{+/-}$ ctrl network only comprised 4 communities (**Figs 6A, 6B, and S7**). None of the communities identified in the WT ctrl group were identical to those of the $Tbr1^{+/-}$ ctrl group, no matter whether brain regions or mean correlation within or between communities was being considered (**Figs 6A, 6B, S6B, and S7 and S3 Data**).

Under BLA activation, the communities in the WT and $Tbr1^{+/-}$ mice were substantially reorganized. All major communities of the WT TBS network (communities 1 to 3, total 613/618 regions) displayed weaker mean correlations within and between communities relative to the first 3 communities of the WT ctrl network (**Figs 6A, 6B, and S7 and S3 Data**, WT TBS communities 1 to 3 versus WT ctrl communities 1 to 3). For the $Tbr1^{+/-}$ TBS network, the brain regions were highly concentrated in 2 communities, i.e., communities 1 and 3, with strong mean within and between community correlations (**Fig 6A and 6B**). Note that community 1 of the $Tbr1^{+/-}$ TBS network is similar to community 2 of the WT ctrl network (**Fig 6B**), suggesting that amygdala activation may reorganize the neural network of $Tbr1^{+/-}$ mice to be more comparable to that in WT mice.

Overall, the community analysis suggests that Tbr1 haploinsufficiency reorganizes how the mouse brain is networked and also leads to differential connectivity upon BLA activation. Importantly, BLA activation renders the neural network of $Tbr1^{+/-}$ mice more similar (to a certain extent) to that of WT mice.

### *Tbr1* deficiency and BLA activation influence the default mode network

Although the above-described community analysis reveals the altered modular integration caused by *Tbr1* deficiency and BLA activation, the physiological functions of the identified communities were unclear. Therefore, we applied our dataset to a physiologically relevant sub-network analysis to explore any differences among experimental groups. Our control groups reflect a cognitive state where the mice were in their home cages during wakefulness and in the absence of any salient stimuli. Therefore, we adopted the default mode network (DMN) as appropriate for our analysis since it was initially identified by resting-state fMRI (functional magnetic resonance imaging) as a set of brain regions with correlated activities in the absence of goal-directed tasks and inactivation during task execution [31]. Given that the DMN has been implicated in social behaviors and ASD [32,33], we posited that our DMN analysis would likely reveal the physiological relevance of the deficits in $Tbr1^{+/-}$ mice.

The core DMN regions in the mouse brain are located in the isocortical prefrontal (ACAd, ACAv, PL, ILA, ORBl, ORBm, ORBvl), somatomotor (SSp-tr, SSp-ll, MOs), and medial (VISa,

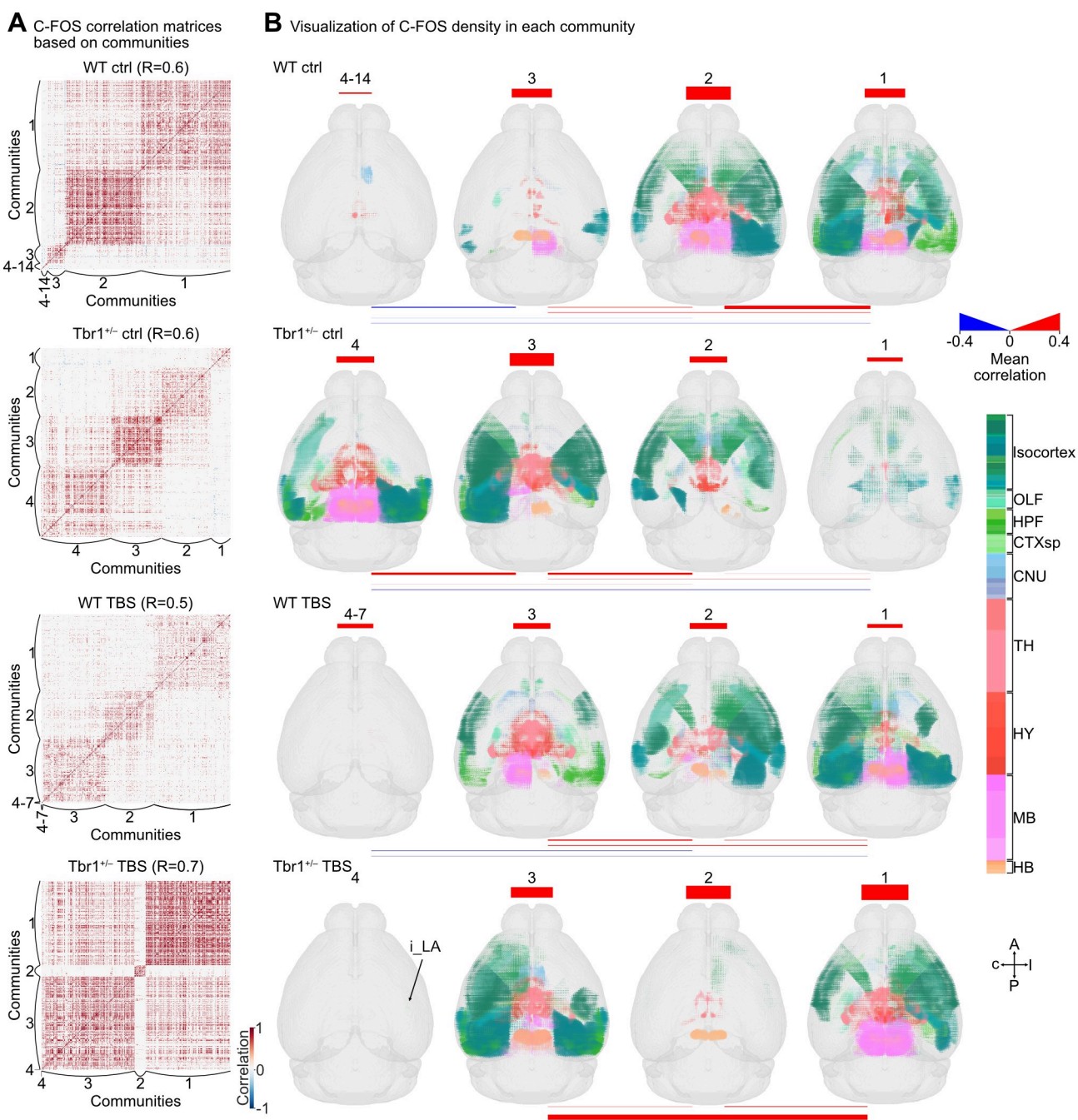

**Fig 6. Modular organization of C-FOS correlation networks.** (**A**) The same datasets of interregional correlations for C-FOS expression are shown in **Fig 5A**, except the interregional pair positions were sorted by their communities. The R values that generate the maximum difference in Q between real and shuffled networks for each experimental group are indicated, and they were used to analyze network communities (also see **S6A Fig**). (**B**) The 3D maps of binned and averaged C-FOS$^+$ cubes for different communities (columns) of the 4 experimental groups (rows). C-FOS$^+$ cubes are shown in brain region color codes, with their sizes representing the relative normalized density of C-FOS$^+$ cells within binned cubes (100 μm$^3$) (also see **S7 Fig**). For better visualization of minor communities of the WT ctrl (4–14) and WT TBS (4–7) groups, the C-FOS correlation matrices and C-FOS$^+$ cube maps (only cube identities and locations) of those communities are also presented in **S6B Fig**. Mean correlations within (the bars at the top of 3D maps) and between communities (the lines under the 3D maps) are annotated according to differential extents (thickness) and polarity (color). Thicker bars or lines represent a higher degree of correlation, whether positive (red) or negative (blue). Sample sizes are the same as in **Fig 5**. The numerical value data are available in **S3 Data**. TBS, theta-burst stimulation; WT, wild-type.

VISam, RSPagl, RSPd, RSPv) areas of both brain hemispheres [34] (**Fig 7A**, right). We extracted our data on oChIEF patterns, C-FOS expression, and functional connectivity for these core DMN regions and compared them among our experimental groups. To aid visualization, the summarized signals in 10 and 100 mm$^3$ cubes were used to highlight differences in oChIEF-Citrine and C-FOS expression, respectively.

Compared to WT mice, *Tbr1*$^{+/-}$ mice exhibited more ipsilateral BLA projections to the DMN regions, except RSPv, RSPd, ORBm, and VISa (**Figs 7A, S8A, and S8B**), implying a role for TBR1 in controlling BLA axonal projections into these regions. For C-FOS expression, we observed higher C-FOS$^+$ cell numbers in bilateral DMN regions for the *Tbr1*$^{+/-}$ ctrl versus WT ctrl and WT TBS versus WT ctrl comparisons (**Figs 7B, S8C, and S8D**). In contrast, there were more DMN regions with reduced numbers of C-FOS$^+$ cells for the *Tbr1*$^{+/-}$ TBS veresus *Tbr1*$^{+/-}$ ctrl comparison (**Figs 7B, S8C, S8D**). Thus, DMN activity is indeed sensitive to *Tbr1* deficiency and BLA activation.

Next, we determined the DC and participation coefficient (PC) of all DMN regions. Many regions of the *Tbr1*$^{+/-}$ ctrl DMN exhibited reduced DC, and only the ipsilateral ACAv (i_ACAv) presented a slightly elevated DC (**Fig 7C**). PC reflects the relative distribution of a given brain region's (node) correlations (edges) with other communities versus the correlations within its community. We found that in contrast to our observations for DC, *Tbr1*$^{+/-}$ ctrl DMN regions showed elevated PC values for many regions relative to the WT ctrl DMN (**Fig 7C**). These analyses again evidence that the connectivity of the DMN in *Tbr1*$^{+/-}$ ctrl mice is abnormal, thus resembling the aberrant DMN displayed by ASD patients [35,36].

We also observed that BLA activation elicited alterations to the DMN in both WT and *Tbr1*$^{+/-}$ mice, albeit with different consequences (**Fig 7C**). In WT mice, BLA activation induced the same decreasing tendency for DC in many of the DMN regions, but various changes in PC (**Fig 7C**, WT TBS versus WT ctrl). In *Tbr1*$^{+/-}$ mice, the same stimulation treatment elevated DC but reduced PC in many of the DMN regions (**Fig 7C**, *Tbr1*$^{+/-}$ TBS versus *Tbr1*$^{+/-}$ ctrl). These differences in the DMN between WT and *Tbr1*$^{+/-}$ mice upon BLA activation demonstrate the differential responses of DMN regions to BLA activation.

Finally, we examined the reciprocal C-FOS correlation within different DMN regions to visualize the interconnected networks of the specific DMN regions (**Fig 7D**). Consistent with a previous study on the mesoscopic mouse DMN [34], our WT mice exhibited a strong interconnected network among DMN regions (**Fig 7D**). However, the DMN of awake *Tbr1*$^{+/-}$ mice (*Tbr1*$^{+/-}$ ctrl) had a much more loosely connected network, showing overall hypoconnectivity among DMN regions (**Fig 7D**). TBS at BLA again exerted opposing effects on *Tbr1*$^{+/-}$ mice and their WT littermates, with the WT TBS group having a loosely connected network, whereas the interconnected network of the *Tbr1*$^{+/-}$ TBS group became stronger than that of *Tbr1*$^{+/-}$ ctrl mice (**Fig 7D**).

Thus, using DMN as an example, our analyses support that *Tbr1* deficiency and BLA activation regulate the connectivity of a brain subnetwork relevant to social behaviors and ASD [32,33].

## Alteration of *Tbr1*$^{+/-}$ mouse social behaviors by BLA activation

Our previous study indicated that BLA deficiency is critical to the autism-linked behavioral deficits exhibited by *Tbr1*$^{+/-}$ mice [17]. Our above-described results from the current study further indicate that TBS at the BLA enhances whole-brain synchronization in *Tbr1*$^{+/-}$ mice so that it is more comparable to that of WT mice. If whole-brain synchronization is indeed a critical aspect of autism-linked deficits, we speculate that TBS at the BLA may improve to some

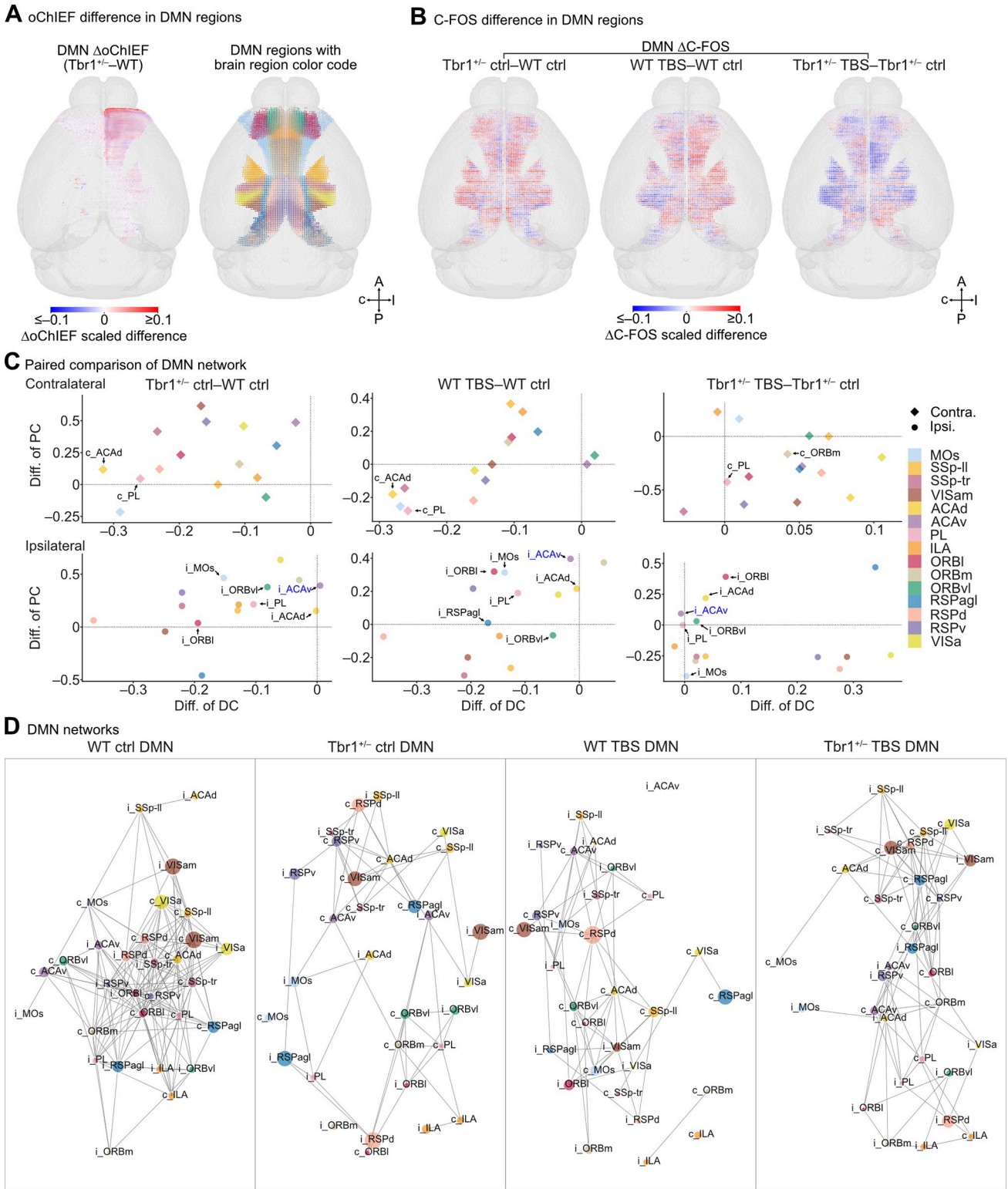

**Fig 7. Comparison of DMNs in WT ctrl, *Tbr1*+/− ctrl, WT TBS, and *Tbr1*+/− TBS groups.** (**A**) Left, the differences of BLA axonal projections into DMN regions for *Tbr1*+/− and WT mice, as revealed by ΔoChIEF for 10 μm³ binned cubes. Right, the colored spots indicate DMN brain region identities. Spot locations are derived from the locations of WT ctrl C-FOS cubes (100 μm³ bins) independent of information on C-FOS⁺ cell density. (**B**) The 3D maps showing differences in density of C-FOS⁺ cells (ΔC-FOS, 100 μm³ bins) for the DMN of experimental group pairs. The heatmaps indicate the scaled differences for ΔoChIEF (**A**) and ΔC-FOS (**B**). (**C**) Structure-wise comparison of DMN properties for 4 experimental group pairs. Each scatterplot shows

the difference in DC versus the difference in PC in each contralateral (top) and ipsilateral (bottom) DMN brain region for each group pair. (**D**) Visualization in spring layout of the DMN networks for the 4 experimental groups. Brain regions (nodes) having many reciprocal C-FOS correlations (edges) are closer together. The color of each node represents brain region identity. The size of the node represents the normalized C-FOS density of the brain region. Labels "i_" and "c_" represent ipsilateral or contralateral brain regions, respectively. DMN region color codes are identical for (**A**), (**C**), and (**D**). In (**C**), the DMN regions indicated by arrows represent those regions showing a significant difference in BLA-derived circuits and C-FOS⁺ cell numbers between pairwise-compared groups (also see **S8 Fig**). The i_ACAv with blue font exhibited a negative correlation between the amount of BLA-derived axonal innervations and its modulatory effects on C-FOS expression patterns. Thus, the i_ACAv received enhanced BLA axonal projections that induce net inhibitory effects on neuronal activities in the *Tbr1*^+/− mouse brain. All DMN regions are ROI-corrected. Sample sizes for the DMN network analysis are identical to those described in **Fig 5**. Sample sizes for the DMN ΔoChIEF and DMN ΔC-FOS analyses are the same as those described in **Fig 4**. The numerical value data are available in **S3 Data**. BLA, basolateral amygdala; DC, degree centrality; DMN, default mode network; PC, participation coefficient; ROI, region of interest; TBS, theta-burst stimulation; WT, wild-type.

extent the social behaviors of *Tbr1*^+/− mice. To investigate that possibility, we set up a paradigm in which the environmental conditions were close to those of our whole-brain circuit analysis to evaluate the effect of TBS at BLA on the social behaviors of *Tbr1*^+/− and WT mice (**Fig 8A**). After unilaterally injecting AAV into BLA, mice were first habituated to experimental handling for 3 days and then went through 2 sets of tests in 2 consecutive days. Each set of tests comprised 4 sessions, i.e., pretreatment, control treatment at Day 1 (D1) versus TBS at D2, posttreatment, and a reciprocal social interaction (RSI) test (**Fig 8A**). Therefore, we could compare the behavioral features of the same mice before and after TBS at BLA.

We applied DeepLabCut together with DeepOF to characterize in detail behavioral features of the mice, including both social and nonsocial behaviors, in different sessions. We found that TBS at BLA noticeably increased the time of nose-to-nose interactions of *Tbr1*^+/− mice with unfamiliar mice in the RSI test. In contrast, the time of nose-to-tail and following behaviors of *Tbr1*^+/− mice were reduced after BLA activation (**Fig 8B**). Although nose-to-nose, nose-to-tail, and following behaviors all represent rodent social behaviors, their meanings are slightly different. Nose-to-tail and following behaviors are more relevant to mating behaviors. However, nose-to-nose interactions, involving social facial touching, are critical for mice to discriminate individual conspecifics [37]. Whisker-dependent somatosensory responses, olfaction, vision, and hippocampus-dependent memory are all involved in the process of social recognition and memory inherent to nose-to-nose interactions. Consistently, our analyses revealed that connections between the BLA and visual and somatosensory cortices and hippocampus were affected by *Tbr1* deficiency and TBS at BLA (**Fig 3 and S1 Data**). Thus, TBS at BLA alters both functional connectivity and patterns of social behavior of *Tbr1*^+/− mice.

In contrast to the effect observed for *Tbr1*^+/− mice, TBS at the BLA did not influence nose-to-nose, nose-to-tail, and following behaviors in WT mice, although it did reduce the time spent in side-by-side contact (**Fig 8B**). Thus, the effect of TBS at BLA specifically impacted the behaviors of *Tbr1*^+/− mice. In terms of nonsocial behaviors, only "looking around" was consistently found to differ in our comparisons (**Figs 8C, S9, and S10**). In particular, *Tbr1*^+/− mice spent more time looking around than WT during the posttreatment period (**S10 Fig**, lowest panel) and the RSI test on D2 (**Fig 8C**). Perhaps BLA activation in *Tbr1*^+/− mice also increases their awareness of the environment, as well as alters their social behaviors.

In conclusion, our detailed behavioral analyses are consistent with the conclusion from our connectivity studies that TBS at BLA alters whole-brain synchronization and social behavior patterns of *Tbr1*^+/− mice.

## Discussion

In the current report, we used a whole-brain mapping and quantification platform to investigate the structural and functional rewiring of BLA-derived circuits in a *Tbr1* ASD mouse model. We conclude that *Tbr1* deficiency reduces contralateral projections of BLA to the

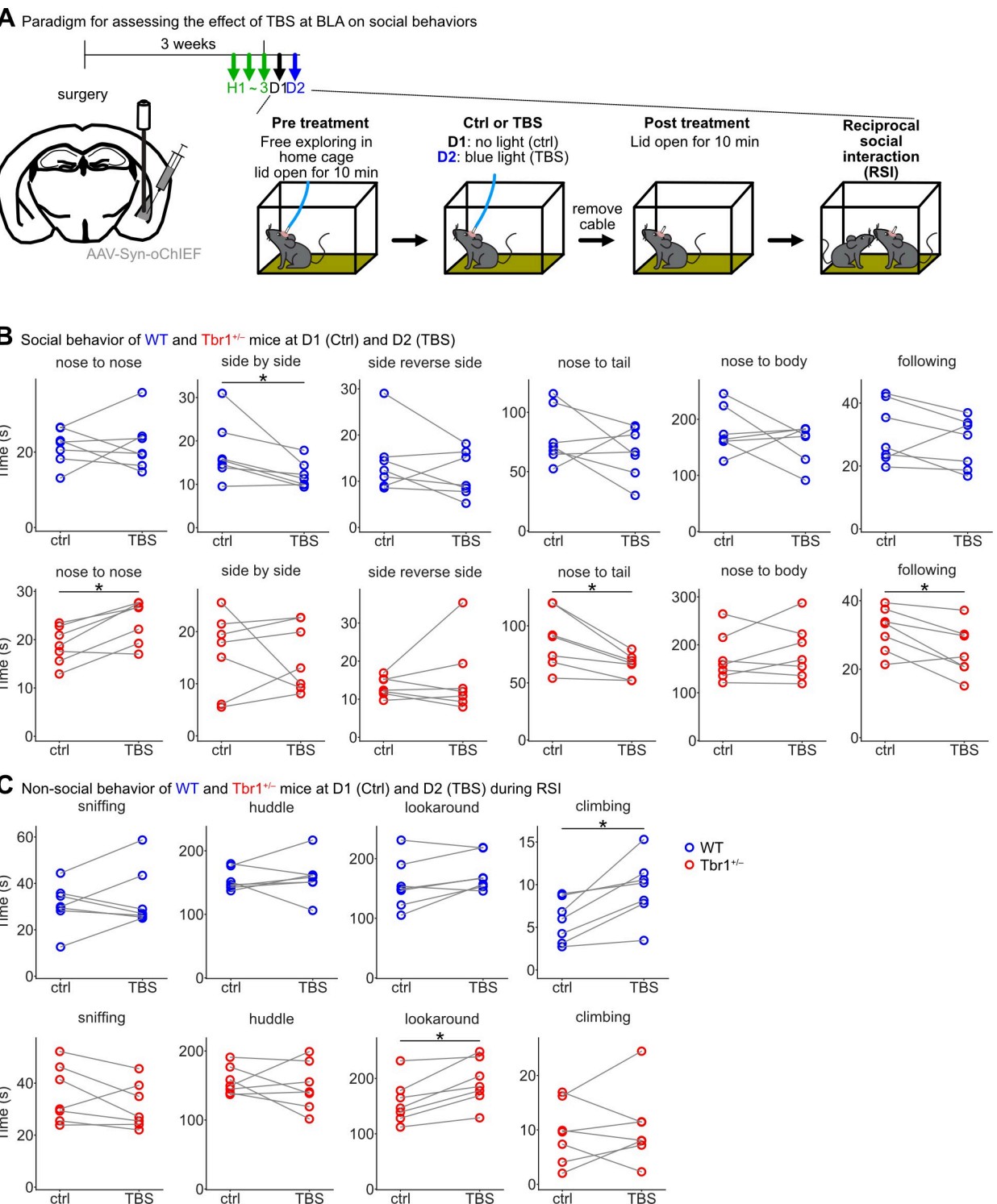

**Fig 8. Changes in behavior induced by TBS at BLA in WT and *Tbr1*[+/−] mice.** (**A**) Experimental timeline. After AAV injection into the BLA, 3 days of habitation were first conducted, followed by 2 sets of tests on 2 consecutive days. Each set comprised 4 sessions: (1) pretreatment; (2) sham treatment (ctrl) or TBS at BLA; (3) posttreatment; and (4) RSI test. (**B**) Comparison of social behaviors before and after TBS at BLA (ctrl vs. TBS). Top panel: results for WT mice (blue circle; nose to nose, $P = 0.9375$, $W = 13$; side by side, $P = 0.03125$, $W = 1$; side reverse side, $P = 0.296875$, $W = 7$; nose to tail, $P = 0.375$, $W = 8$; nose to body, $P = 0.578125$, $W = 10$; following, $P = 0.296875$, $W = 7$). Bottom panel: results for *Tbr1*[+/−] mice (red circle; nose to nose, $P = 0.03125$, $W = 1$; side by side, $P = 0.8125$, $W = 12$; side reverse side, $P = 0.8125$, $W = 12$; nose to tail, $P = 0.015625$, $W = 0$; nose to body, $P = 0.6875$, $W = 11$; following, $P = 0.046875$, $W = 2$). (**C**) Comparison of nonsocial behaviors during the RSI session before

and after TBS at BLA (ctrl vs. TBS). Top panel: results for WT mice (blue circle; sniffing, $P = 0.578125$, $W = 10$; huddle, $P = 0.6875$, $W = 11$; looking around (lookaround), $P = 0.078125$, $W = 3$; climbing, $P = 0.015625$, $W = 0$). Bottom panel: results for *Tbr1*$^{+/-}$ mice (red circle; sniffing, $P = 0.21875$, $W = 6$; huddle, $P = 0.578125$, $W = 10$; lookaround, $P = 0.015625$, $W = 0$; climbing, $P = 0.578125$, $W = 10$). Sample sizes of mice: WT, $n = 7$; *Tbr1*$^{+/-}$, $n = 7$. Two-tailed Wilcoxon signed-rank test was used for statistical analyses. *$P < 0.05$. The numerical value data are available in **S4 Data**. AAV, adeno-associated virus; BLA, basolateral amygdala; RSI, reciprocal social interaction; TBS, theta-burst stimulation; WT, wild-type.

forebrain and results in widespread mistargeting of BLA axons to ipsilateral hemispheres and contralateral TH, HY, and MB. Unexpectedly, neuronal activity is generally up-regulated in *Tbr1*$^{+/-}$ mice in the absence of specific stimulation, but it is noticeably reduced when TBS at BLA was applied to *Tbr1*$^{+/-}$ mice, indicating that *Tbr1* deficiency alters whole-brain connectivity. Based on the interregional correlation of C-FOS$^{+}$ cell density, we have further shown that *Tbr1* deficiency reduces the synchronization of interregional neuronal activities and elicits modular reorganization of the C-FOS correlation network. Moreover, TBS at BLA greatly enhances interregional synchronization of *Tbr1*$^{+/-}$ mice at a whole-brain scale. Furthermore, TBS at BLA increases the activity correlation, i.e., degree of synchronization, of the DMN, a subnetwork that has been linked to social behaviors and ASD [32,33]. TBS treatment also increased the time of nose-to-nose social discrimination behaviors of *Tbr1*$^{+/-}$ mice. Thus, our study reveals unexpected changes in brain connectivity attributable to *Tbr1* deficiency and further illuminates a potential treatment for TBR1-related autism by means of deep brain stimulation at the amygdala. Note that we conducted unilateral stimulation at BLA in the current study. It is plausible that bilateral stimulation also exerts a beneficial effect, although further investigation would be required to confirm that speculation.

The BLA is the most sensitive brain region to *Tbr1* haploinsufficiency [17]. However, TBR1, a well-known marker of glutamatergic projection neurons in the forebrain, is also expressed in the cerebral cortex, amygdala, olfactory bulb, and hippocampus [15,17,22,38–40]. Thus, when we investigated the C-FOS$^{+}$ cell density in the control group without any specific stimulation, it was unexpected that *Tbr1*$^{+/-}$ mice generally had more C-FOS$^{+}$ cells than WT mice. Given that specific knockout of *Tbr1* from layer 6 neurons of the cerebral cortex results in a reduced density of synapses for both projection neurons and interneurons in the cerebral cortex [41], it seems likely that *Tbr1* deficiency also indirectly influences the development and function of local interneurons and locally leads to abnormal neuronal activation. Activated projection neurons then control neuronal activity in other brain regions and result in the generally higher neuronal activation displayed by *Tbr1*$^{+/-}$ mice. Notably, after TBS at BLA, WT and *Tbr1*$^{+/-}$ mice exhibited opposing responses. No matter whether the regions had more or less BLA-derived axon innervation, TBS treatment at BLA tended to reduce the neuronal activity of *Tbr1*$^{+/-}$ mice. This net effect of BLA mistargeting may be caused by both direct and indirect connections controlled by the BLA, as well as feed-forward or feedback inhibitions mediated by local microcircuits. Nevertheless, these unexpected findings strengthen the necessity for in vivo whole-brain analysis given that neural connections and their consequent effects on the brain are exceedingly complex. In terms of studying neural dysconnectivity syndromes such as ASD, our unbiased structural and functional whole-brain analyses based on histological axon labeling and C-FOS expression represent a comprehensive screen of candidate brain regions for further investigation.

The TBS protocol that we applied to mice likely triggers plasticity, such as long-term potentiation or long-term depression, and leads to a long-term effect on neural activation and mouse behaviors. We do not know exactly how long the changes in plasticity might be maintained. However, the onset of behavioral changes after TBS at BLA appears to be rapid because we observed differences in social behaviors when we executed the RSI test 10 min after TBS at

BLA. Further investigations need to be conducted in the future to understand how long the effect of TBS at BLA is maintained.

We observed that whole-brain synchronization of $Tbr1^{+/-}$ mice was enhanced by TBS at BLA, even though their C-FOS$^+$ cell density was reduced. This treatment further increased the nose-to-nose investigations of $Tbr1^{+/-}$ mice and their awareness of the environment, supporting the physiological relevance of the increased whole-brain synchronization elicited by TBS at BLA. This treatment strategy may correct neural connectivity by modulating the plasticity of indirect BLA innervations and local microcircuits to achieve enhanced whole-brain synchronization. The changes in connectivity and behavioral improvements afforded by TBS treatment at BLA described herein provide a potential framework for the deep brain stimulation proposed previously as a potential therapeutic strategy for ASD patients [42–44], especially for stimulations focused on the amygdala [45,46]. It would be interesting to further generalize the role of the amygdala in whole-brain synchronization and autism-linked behaviors in other mouse genetic models.

In our current study, we used a pan-neuronal promoter to express oChIEF in BLA and explored all possible alterations of BLA-derived circuits. TBS at BLA can activate all oChIEF-expressing BLA neurons. This stimulation paradigm mimics the deep brain stimulation applied to human patients, representing a protocol that cannot target specific neuron types. Nevertheless, it is worth considering cell-specific options in the future to understand the relevance between function and circuitry of the BLA. Using TBS to activate specific types of cells or specific axonal projecting neurons in the BLA would further elucidate the specific connectivities and related functions of the BLA, helping to establish how different BLA cell types and circuits regulate different behaviors controlled by this brain region.

In addition to whole-brain connectivity deficits, we have also shown that *Tbr1* deficiency alters the structural and functional connectivity between the BLA and DMN, a specific subnetwork relevant to social deficits and ASD [32,33]. A previous study indicated that regions within the mouse DMN are structurally and functionally connected and that intercorrelated functional DMN connectivity is related to structural connectivity [34]. An abnormal DMN results in atypical information integration and prompts social-cognitive deficits [33,47]. Unlike in humans, there are much fewer studies on the DMN in ASD mouse models [48,49]. To our knowledge, the current study provides the first evidence linking the DMN with the amygdala, a critical brain region for social behaviors, in a mouse genetic model. Our findings reveal that BLA activation results in opposing consequences for $Tbr1^{+/-}$ mice and WT littermates in terms of neuronal activity and internetwork connectivity of the DMN. Similar to global brain synchronization, TBS-induced BLA activation elicits stronger connectivity among regions of the DMN in $Tbr1^{+/-}$ mice, thereby likely improving DMN function. Thus, TBS at BLA may affect DMN regions to alter the social behaviors of Tbr1$^{+/-}$ mice. In human brains, the amygdala also projects to the DMN [50]. It is intriguing to explore if deep brain stimulation of the amygdala of ASD patients also enhances the connectivity and synchronization of their DMN, thereby improving their symptoms.

Our previous studies have demonstrated that only the anterior parts of the lateral and basolateral amygdala (i.e., the LA and BLA) in the 2 brain hemispheres directly innervate to and reciprocally up-regulate the neural activity of each other via the posterior part of the anterior commissure [17,23]. *Tbr1* deficiency impairs contralateral axonal projections of the BLA [17,23]. Herein, we have confirmed that axonal projection of the BLA of a brain hemisphere in $Tbr1^{+/-}$ mice fails to target the contralateral BLA (**Fig 2**) and that the anterior part of the BLA was ready to be activated by the contralateral BLA in WT mice but not $Tbr1^{+/-}$ mice (**Fig 4B**, BLA, bottom). Note that when we summed C-FOS signals across the entire LA and BLA, only the LA exhibited a difference between WT TBS and WT ctrl. For the BLA, statistical analysis

did not reveal a difference (**S1 Data**). The unchanged posterior part of the BLA likely masks the difference in the anterior part (**Fig 4B**, BLA, bottom). These results highlight the sensitivity of slice-based quantifications and the necessity to analyze signals along the AP axis, a significant strength of our analytical platform.

## Methods

### Ethics statement

All animal experiments were performed with the approval of the Academia Sinica Institutional Animal Care and Utilization Committee (Protocol No. 14-11-759 and 18-10-1234) and in strict accordance with its guidelines and those of the Council of Agriculture Guidebook for the Care and Use of Laboratory Animals. Human subject study is not included in the current work.

### Mice

The *Tbr1*$^{+/-}$ mice were originally provided by Drs. J. L. Rubenstein (Department of Psychiatry, University of California, San Francisco) and R. F. Hevner (Department of Neurological Surgery, University of Washington, Seattle) and had been backcrossed to WT C57BL/6 for more than 40 generations. *Tbr1*$^{+/-}$ mice and WT littermates were housed in the animal facility of the Institute of Molecular Biology, Academia Sinica, under controlled temperature and humidity and a 12-h light/12-h dark cycle. All animal experiments were performed at 3 to 5 mo of age with the approval of the Academia Sinica Institutional Animal Care and Utilization Committee and in strict accordance with its guidelines (Protocol No. 14-11-759 and 18-10-1234).

### Virus production

For axon tracing and in vivo stimulation of BLA-derived circuits, we packaged an AAV that expresses the ultrafast ChR variant oChIEF under the control of the human Synapsin promoter (AAV-Syn-oChIEF-Citrine and AAV-Syn-oChIEF-tdTomato) as described in our previous study [23]. Briefly, a transgenic plasmid encoding oChIEF-Citrine, AAV helper plasmid serotype AAV8 capsid and AAV2 replication proteins, together with an adenovirus helper plasmid encoding essential proteins for AAV packaging, were transfected into adenovirus E1-expressing cells. Crude virus soup was obtained from transfected E1-expressing cells and purified by iodixanol gradient centrifugation. The physical vector titers were quantified by measuring the number of packaged vector genomes by real-time PCR using SYBR Green reaction mix (Roche Diagnostics, Mannheim, Germany).

### Experimental procedure of the mesoscopic whole-brain scale circuit analysis pipeline

In order to analyze the structural and functional connectivity of BLA-derived circuits, we developed a mesoscopic whole-brain scale circuit analysis pipeline (**Figs 1** and **S1**). Detailed methods are described in the following sections.

**Step1: Brain sample preparation.** To label the BLA-derived circuits and manipulate their activity for examining functional consequences, we performed stereotaxic surgeries for virus infection and optic fiber implantation. Mice were deeply anesthetized and their heads were secured on a Lab Standard Stereotaxic Instrument (Stoelting, Wood Dale, IL USA). AAV-Syn-oChIEF-Citrine (0.1 μl, $10^{10}$ genomes/μl) was slowly infused over 10 min unilaterally into the BLA (virus injection coordinates: 1.3 mm posterior, 3.3 mm lateral, and 4.95 mm ventral to the bregma). Upon completing virus injection, an optic fiber was slowly inserted into the site

to 0.05 mm above the virus injection site (optic fiber implantation coordinates: 1.3 mm posterior, 3.3 mm lateral, and 4.9 mm ventral to the bregma). All injection sites are listed in **S2A Fig**. Based on the Allen Mouse CCFv3, the AAV injection sites covered the LA, BLAa, and BLAp, which represent BLA in our previous publications [17,23]. To make our findings consistent with those of our previous studies, we still use BLA to represent a combination of the LA, BLAa, and BLAp when we mention the AAV infection site. Thus, it is slightly different from the meaning of BLA in CCFv3, which represents the regions containing the BLAa, BLAp, and BLAv and excludes the LA.

After recovering for at least 3 weeks, the mice were habituated to being handheld, to having a patch cable connected to the optic fiber, and to the lid of their home cage being opened for 10 min when the patch cable was being connected. Habituation sessions were conducted once per day for 3 d. In vivo optogenetic stimulation and brain sample fixation were carried out on the day following the last habituation session. We used a 473-nm laser with 10 mW light intensity (emitted from the tip of the optic fiber) for optogenetic stimulation. After 10 min of freely exploring their home cage, mice received 5 repetitive theta burst frequency stimulations (TBS: 10 trains/TBS, intertrain interval, 200 ms; 5 pulses/train, interpulse interval, 10 ms) with a 30-s inter-TBS interval to a unilateral BLA. This experimental stimulation paradigm mimics physiological theta oscillations and has been demonstrated to intensify responses to cortical and thalamic inputs through commissural BLA projections [23]. TBS was also used to induce changes in the neuronal plasticity of the BLA [51]. Therefore, the functional consequences of TBS on BLA-derived circuits can persist and prevail over the random activation of BLA neurons during wakefulness. After photostimulation or sham manipulation (without light emission), mice were kept in their home cage for 2 h until killed by perfusing them with 4% paraformaldehyde in PBS. The brains were then dissected out and postfixed overnight at 4˚C.

**Step2: High-content immunofluorescence imaging.** Brains were cryopreserved in 30% sucrose at 4˚C for 2 d and then embedded in OCT (4583, Tissue-TeK). For whole-brain analysis, we used a cryostat microtome (CM1900, Leica) to cryosection each brain from the posterior to anterior cerebrum into at least 120 consecutive coronal brain slices (60 μm/slice), with the cerebellum and olfactory bulb being excluded from our analysis. Traditional immunohistochemical methods were deployed, as described previously [23]. Primary anti-GFP (1:5,000, Abcam) and anti-C-FOS (1:200, Cell Signaling) antibodies were used to reveal Citrine-containing BLA axonal projections and C-FOS immunoreactivities, respectively. The stained coronal brain sections (at least 120 sections) with intensified oChIEF-Citrine (stained with GFP antibody) and C-FOS (stained with C-FOS antibody) signals were imaged with a high-content imaging system (Molecular Devices) hosting a 4× objective. Images from one brain were stitched together using MetaXpress 6.2.3.733 (Molecular Devices) and manually aligned in Amira 6.4 (Thermo Fisher Scientific) or Etomo [52].

**Step3: Registration and transformation to Allen Mouse Common Coordinate Framework version 3 (CCFv3).** After acquiring the aligned consecutive slice raw images, we registered the image series to CCFv3 [25]. We downloaded high-resolution coronal templates with a 10-μm resolution along the AP axis using Allen Software Development Kit Python code and then manually matched the corresponding template to each raw image according to certain landmark brain structures, such as the beginning and end of the hippocampus (HIP), the location of the anterior commissure (act and aco), and the disconnections of the corpus callosum (cc). After determining the matched template for each raw image, binary experimental images were registered to the matched binary templates by means of the medical image registration library SimpleElastix [53], first with 12 global (translation and affine) and then 1 local (B-spline) transforming parameters. Finally, the stored multiple transforming parameters were

applied to merge the experimental images of oChIEF-Citrine and C-FOS signals into registered images.

**Step4: Image segmentation for oChIEF and C-FOS signals (also see S1 Fig).** Before quantification of oChIEF and C-FOS signals and comparing the differences between WT and $Tbr1^{+/-}$ mice, we needed to eliminate fluorescence signals due to experimental variation and background interference. The purpose of image segmentation for oChIEF signals was to determine which pixels are oChIEF-positive (**S1A Fig**). Therefore, first we deployed the pixel classification algorithm ilastik1.3.3 [54] to predict the probability that every pixel in a registered image displayed oChIEF signals. The oChIEF probability maps of each image were cross-referenced against the registered images pixel by pixel. The resulting filtered images (raw signal intensity × probability) were further processed to intensify edges by subtracting the Gaussian blurred version of the filtered images. The edge-enhanced filtered images were then subjected to thresholding/binarization ($>6 \times$ S.D. of the Z-stacked image with filtering and edge enhancement) to isolate oChIEF-positive pixels.

The aim of image segmentation for C-FOS signals was to determine the locations of C-FOS$^+$ cells within each image (**S1B Fig**). We preprocessed the raw C-FOS images by applying rolling ball background subtraction (ImageJ, radius 50 pixels). The spot detection algorithm of Imaris 9.3 (Bitplane) was then applied to detect C-FOS$^+$ cells from background-subtracted images using consistent criteria (detection spot diameter: 10 μm; detection threshold: $2 \times$ S.D. of the Z-stacked image with background subtraction). The locations of C-FOS$^+$ cells in each image could then be exported. Finally, the binarized oChIEF signals and C-FOS$^+$ cell locations of each registered image were subjected to quantification.

**Step5: Application of brain structure masks and quantification of oChIEF and C-FOS signals.** We accessed the Reference Space of CCFv3 to obtain the coronal indicator masks of each brain region by using Allen Software Development Kit Python code. The coronal indicator masks could separately compartmentalize oChIEF-positive pixels and C-FOS$^+$ cells into different brain regions, which served as regions of interest (ROIs) to quantify the binarized oChIEF pixel number or C-FOS$^+$ cell number for each brain region along the AP axis. All computations were carried out using the Numpy, Pandas, OpenCV, scikit-image, and Geo-Pandas Python packages. The data compartmentalizing oChIEF-positive pixels and C-FOS$^+$ cells into different brain regions were subsequently used for 3D visualization. Data on quantified binarized oChIEF pixel number, C-FOS$^+$ cell numbers, and brain region area were used for the analysis. Some brain regions were excluded from the analysis of oChIEF signals, such as ipsilateral BLA-related regions (LA, BLA, BLAa, BLAp) and bilateral VISam, which contained artificial signals caused by experimental surgery for virus injections. Moreover, the bilateral somatosensory cortex (SSp, SSp-n, SSp-bfd, SSp-ll, SSp-tr, SSp-un, SSs) and corpus callosum-related fiber tracts (cc, fa, ec, ee, ccb, ccs) were also excluded from the analysis due to exiguous oChIEF-Citrine-expressing AAV infections during surgery.

## k-nearest neighbors (KNN) algorithm-based classification

To examine the spatial separability of virus injection sites, we conducted a KNN-based classification with 5-fold cross-validation using the scikit-learn Python package. A series of numbers of nearest neighbors was used to identify the one achieving the best KNN classification accuracy. Furthermore, a randomized dataset of virus-injecting locations or PC locations, shuffled 1,000 times, between experimental groups was used to validate the significance of spatial separability. The spatial separability was considered significant when the KNN classification accuracy of the real dataset deviated from the same metric distribution calculated for the randomized dataset ($P < 0.05$).

## Slice-based quantification and related analysis

Slice-based quantifications of oChIEF and C-FOS signals involved calculating the oChIEF pixel and C-FOS$^+$ cell distributions along the AP axis of each brain region, respectively. To achieve fair comparisons, we carried out 100 μm binning of oChIEF pixel and C-FOS$^+$ cell distributions along the AP axis. The binned oChIEF pixel and C-FOS$^+$ cell distributions were then used for structure-wise comparisons and correlation analysis between rewired BLA-derived circuits and their C-FOS modulatory ability.

**Structure-wise comparison of the data from slice-based quantification (also see S3 Fig).** Before structure-wise comparison of oChIEF pixel or C-FOS$^+$ cell distributions between different groups (oChIEF data, WT and *Tbr1*$^{+/-}$; C-FOS data, WT ctrl, *Tbr1*$^{+/-}$ ctrl, WT TBS, and *Tbr1*$^{+/-}$ TBS), we set a threshold to select brain regions for comparison. To compare oChIEF pixel distributions, brain regions were selected with the summed total oChIEF pixels for both groups (WT + *Tbr1*$^{+/-}$) greater than 13-fold the standard deviation of all oChIEF pixels in all contralateral brain regions and larger than the 20th percentile among all brain regions of the Tbr1$^{+/-}$ group (154 contralateral and 322 ipsilateral regions). To compare C-FOS$^+$ cell distributions, brain regions with a summed total C-FOS$^+$ cells for all compared groups (WT ctrl, *Tbr1*$^{+/-}$ ctrl, WT TBS, and *Tbr1*$^{+/-}$ TBS) greater than 100 cells were selected (313 contralateral and 311 ipsilateral brain regions). For our structure-wise comparisons of oChIEF pixel and C-FOS$^+$ cell distributions along the AP axis, Friedman one-way repeated measure analyses were conducted using the Pingouin Python package. A post hoc nonparametric pairwise test with one-step Bonferroni correction of $P$ value (using Pingouin) was conducted for the C-FOS$^+$ cell distributions if the outcome of the Friedman one-way repeated measure analysis was significant ($P < 0.05$). To express the extent of distributional differences between compared groups, we conducted pairwise subtractions of the oChIEF pixels or C-FOS$^+$ cell numbers for the same AP coordinates between all possible pairs. The differences along the AP axis of all sample pairs were then averaged and summed into single metrics (sum of ΔoChIEF or sum of ΔC-FOS) to represent the extent of any differences.

**Correlation analysis of rewired BLA-derived circuits and their C-FOS modulatory ability.** Brain regions selected for correlation analysis were those receiving differential BLA axonal innervations between WT and *Tbr1*$^{+/-}$ mouse brains (Friedman test $P$ value < 0.05) and altered C-FOS$^+$ cell distributions upon TBS-mediated BLA activation in either WT or *Tbr1*$^{+/-}$ mice (adjusted $P$ value < 0.05 in WT TBS versus WT ctrl or *Tbr1*$^{+/-}$ TBS versus *Tbr1*$^{+/-}$ ctrl). First, we determined ΔoChIEF traces by pairwise subtraction of oChIEF-Citrine signals between *Tbr1*$^{+/-}$ and WT samples. The neural activation induced by TBS in *Tbr1*$^{+/-}$ mice and WT littermates, i.e., WT ΔC-FOS traces and *Tbr1*$^{+/-}$ ΔC-FOS traces, was also determined. Pairwise subtraction between *Tbr1*$^{+/-}$ ΔC-FOS traces and WT ΔC-FOS traces was then executed to obtain all possible differences, i.e., ΔΔC-FOS traces. A Pearson correlation coefficient (r) was calculated between the averaged ΔoChIEF trace and the averaged ΔΔC-FOS trace for each brain region. To determine if any correlation was meaningful, i.e., distinct from noncorrelated pairs, the order of ΔoChIEF traces, WT ΔC-FOS traces, and *Tbr1*$^{+/-}$ ΔC-FOS traces along the AP axis was independently reshuffled 1,000 times to create 1,000 putatively noncorrelated mean ΔoChIEF and mean ΔΔC-FOS shuffled pairs. We defined that a meaningful positive or negative correlation between mean ΔΔoChIEF and ΔΔC-FOS values should fit 3 criteria. First, the r value should deviate from the same metric obtained from the shuffled dataset (r$_{shuffled}$) (positive correlation, above 95% r$_{shuffled}$ distribution; negative correlation, below 95% r$_{shuffled}$ distribution). Second, the r value should be >0.1 (positive correlation) or <−0.1 (negative correlation). Third, the tendency thresholds of ΔoChIEF and ΔΔC-FOS traces were determined according to the mean 99th percentiles of scaled differences, which were calculated

from the pair-subtraction population dataset with 1,000 times AP axis reshuffling (**Fig 4B and 4D**, dashed line in middle panel). A positive or negative correlation between mean ΔoChIEF and ΔΔC-FOS depended on the product of ΔoChIEF and ΔΔC-FOS tendency thresholds (positive correlation, >0; negative correlation, <0). The computations of all correlation analyses were carried out using Pandas and Numpy.

## Network analysis of functional connectivity based on interregional C-FOS correlations

Functional connectivity, as measured in our study, was based on the synchronization of C-FOS expression levels between 2 brain regions in individual subjects. To determine functional connectivity, we first conducted volume-based quantification of the C-FOS$^+$ cell density of each brain region (**S4 Fig**). To calculate the C-FOS density, total C-FOS$^+$ cells within a brain region were calculated by summing over all C-FOS$^+$ cells in individual slices and then dividing by the volume of the brain region (converted to mm$^3$). We excluded brain regions with low C-FOS density (summation of C-FOS density from all samples <800 cells/mm$^3$) for subsequent analytical steps. The relative C-FOS density for a given brain region of each sample was obtained by normalizing it to the averaged value calculated from all WT ctrl samples. Therefore, we used relative C-FOS density as a single metric to represent the C-FOS level of a given brain region of one sample. To measure all possible synchronizations, we evaluated interregional C-FOS correlations between all possible brain region pairs by means of two-sided Kendall's rank correlation (using scipy.stats) across all the samples in the same experimental group. A network of interregional C-FOS correlations was constructed using the significant C-FOS correlations (edges, significance level $P < 0.05$) between brain region pairs (nodes) by using the NetworkX Python package. Self-to-self correlations were removed. The DC and PC of individual nodes (brain regions) were calculated using NetworkX.

To dissect the modular organization of networks, we adopted a Louvain community detection algorithm and the modularity metric (Q) calculation provided by NetworkX. To identify the optimal resolution parameter R value used for Louvain community detection, we generated a shuffled network by maintaining the same correlation weights (edges) while randomizing the source and target regions (nodes). An R value series (0.4 to 0.9 in steps of 0.1) was used for Louvain community detection of real and shuffled networks. For each R value, the difference in Q between the modularity of the real network and the shuffled network was calculated 1,000 times. The most appropriate R value that generated the maximal mean difference of Q (Q–Q$_{shuffled}$) was selected for Louvain community detection. These computations were achieved using Numpy, Pandas, and NetworkX.

## ROI corrections of brain region masks

We selected 44 contralateral and 52 ipsilateral brain regions for manual correction of the mismatched ROI in every single brain slice image of each brain sample. To do this, the original coronal indicator masks of each brain region were separated into contralateral and ipsilateral masks and converted to ImageJ ROIs by sequentially applying the ImageJ Create Selection function and adding it to ImageJ ROI Manager. Using the C-FOS or oChIEF raw images (registered to CCFv3) as references, ROIs were verified and corrected by manually adjusting their shapes and locations. After ROI corrections, the ROIs of each brain region in both hemispheres were converted back to image masks by applying the ImageJ Create Mask function. The new coronal indicator masks were then used for requantifications.

### 3D visualization

**Summarized 3D stack of binarized oChIEF images.** To visualize differences in BLA-derived circuits between WT and $Tbr1^{+/-}$ mice, the binarized oChIEF image series of individual brains was binned on the dorsal-ventral and medial-lateral axis into 10 μm/pixel. Each binned pixel intensity represented the oChIEF probability. The AP coordinates of the binned oChIEF image series were converted into the respective CCFv3 coordinates (10 μm/pixel), and all binned oChIEF image series of individual brains were then merged. If the same AP coordinate contained multiple binned oChIEF image series from different brain samples, we conducted averaging on the intensity of those binned oChIEF image series. If an AP coordinate contained only 1 binned oChIEF image, then that binned image represented the oChIEF signals for that particular AP coordinate. The 3D stack of binned and averaged oChIEF image series was adopted to illustrate BLA-derived circuits at a whole-brain scale. Image processing was carried out using ImageJ, Numpy, and scikit-image.

**3D mapping of binned BLA-derived circuits or C-FOS⁺ cell distributions.** To visualize the innervations of BLA-derived circuits or the distributions of C-FOS⁺ cells in particular brain regions, we used the datasets on compartmentalized oChIEF-positive pixels and C-FOS⁺ cells for different brain regions. The CCFv3 whole-brain shell was obtained from the image stacks of CCFv3 "root" coronal indicator masks and converted to an isosurface mesh with smoothing using the vedo Python package. For BLA-derived circuit visualization, binned and averaged oChIEF image stacks within particular regions were employed and binned pixels (10 μm³ cubes) were converted into vectorized points using Numpy, Pandas, and vedo. Each vectorized point expressed different information, such as the brain region to which the point belonged (discontinuous color map), scaled metrics expressing the difference between compared groups (continuous color map), and oChIEF probability (alpha level). Vectorized oChIEF points were mapped onto the 3D whole-brain shell using vedo. To visualize C-FOS⁺ cell distributions, the locations of C-FOS⁺ cells were first binned by 100 μm³ cubes in every brain sample, and then the averaged C-FOS density of every 100 μm³ cube was calculated across brain samples of the same experimental group. Every binned and averaged C-FOS cube (100 μm³) within particular regions was converted into a vectorized sphere using Numpy, Pandas, and vedo. Each vectorized sphere expressed different information, such as the brain region to which the sphere belonged (discontinuous color map), scaled metrics expressing the difference between compared groups (continuous color map), and the normalized C-FOS density within the cube (size of the sphere). Vectorized C-FOS spheres were mapped onto the 3D whole-brain shell using vedo.

**3D mapping of C-FOS correlation networks among main brain areas.** To visualize the C-FOS correlation networks shown in **Fig 5**, the central locations of the main isocortex modules (prefrontal, somatomotor, medial, lateral, auditory, and visual) and subcortical areas (OLF, HPF, CTXsp, CNU, TH, HY, MB, and HB) in both hemispheres were first determined by averaging the locations of all the binned 100-μm³ cubes belonging to particular brain areas. Mean correlations within and between brain areas were then calculated. Centroids representing particular brain areas in the ipsilateral or contralateral hemisphere were converted to vectorized spheres using vedo. The color of each sphere represented brain area identity and the size reflected mean correlation extent. To visualize inter-area mean correlations, correlation edges were generated using vedo. For each edge, color represented correlational polarity and thickness reflected mean correlation extent. Vectorized centroid spheres (brain areas) and inter-area edges (inter-area mean correlations) were mapped onto a 3D whole brain shell again using vedo.

## Behavior paradigm for examining the effect of TBS at BLA on reciprocal social interaction

To manipulate BLA-derived circuits and examine its functional impacts in terms of behavioral changes, we expressed AAV-Syn-oChIEF-Citrine or AAV-oChIEF-tdTomato in mice and inserted optic fibers in their BLA. Mice were single-housed after surgery. On days 19 to 21 after surgery, we conducted the same habituation sessions as conducted for the mesoscopic connectome experiments (once per day for 3 d). RSI was carried out on the day following the last habituation session. The behavior paradigm comprised 4 sessions. The first session was a pretreatment, which allowed the test mouse with a connected patch cable to freely explore the lid-opened homecage for 10 min. The second session differed between Day 1 (D1) and D2. For D2, the same TBS protocol (473 nm laser with 10 mW light intensity) as applied for the mesoscopic connectome experiment was applied to activate the BLA. For D1, mice underwent sham stimulation (ctrl, no light). The third session, i.e., posttreatment, was for recovery from treatment. The patch cable was removed and the mouse was put back in the lid-opened homecage for an additional 10 min. Finally, RSI was carried out by introducing an unfamiliar and younger (at least 30 d younger) mouse into the homecage. The social interactions of the test mouse with the unfamiliar mouse were recorded for 10 min. Mouse behaviors were continuously recorded from a top-down perspective by a digital camera (30 frames/s) during the entire behavior paradigm.

## Deep learning–based analyses of behavior videos

Single videos of the entire behavior paradigm were cropped, trimmed, and separated into 10 min pretreatment, posttreatment, and RSI videos (each video contains 18,000 frames), respectively, using the FFMPEG ImageJ plugin and OpenCV python package. To achieve more objective and comprehensive behavioral analysis, we used DeepLabCut [55,56] for pose estimation and then applied DeepOF [57] to annotate the behaviors in video frames. Single-animal and multi-animal modes of DeepLabCut 2.3.7 were used for pose estimation of pretreatment/posttreatment videos and RSI videos, respectively. According to the DeepOF analysis pipeline, 11 body parts per mouse were annotated, including the nose, right and left ears, 4 limbs, and 3 points along the spine and tail base. DeepLabCut models for single- and 2-animals were trained separately according to the DeepLabCut workflow, which includes extracting frames for labeling, creating a training dataset, training the network, evaluating the trained network, applying the trained model for videos, and relabeling outlier frames for repeated training of models. After repeating the DeepLabCut workflow 3 times and achieving suitable pose estimation results, DeepLabCut models were applied to the videos to obtain the tracklet datasets of all body parts. In addition, tracklets associated with RSI videos were manually verified and corrected for identity shift and incorrectly annotated body parts before subsequent behavioral annotation.

For behavioral annotations, we used the supervised pipeline of DeepOF v0.6 to label social and nonsocial behaviors in each frame. All the predefined rules and classifier from DeepOF were directly used for analysis with one modification, i.e., the threshold distance of the nose to other body parts was restricted to 15 mm. For each social and nonsocial behavior, total behavior time was calculated as the sum of all labeled behavior frames and converted to seconds. Pairwise comparisons of behavior time between groups were conducted (ctrl versus TBS, pretreatment versus posttreatment). Two-tailed Wilcoxon signed-rank test (using Pingouin) was used for statistical analysis.

## Quantification and statistical analysis

Sample size n represents the number of brain samples from individual animals or number of animals in the behavior assay for each experimental group. We did not use statistical methods to predetermine sample sizes, but our sample size is similar to that of a previously published brain connectome study [58]. KNN-based classification of virus injection sites, slice-based quantifications of oChIEF and C-FOS signals, and the $\Delta$oChIEF/$\Delta\Delta$C-FOS correlation analysis are based on $n = 13$ WT (6 from WT ctrl and 7 from WT TBS groups) and $n = 14$ *Tbr1*$^{+/-}$ brain samples (7 from *Tbr1*$^{+/-}$ ctrl and 7 from *Tbr1*$^{+/-}$ TBS groups). Volume-based quantification of relative C-FOS$^+$ cell density and interregional C-FOS correlation analysis was based on samples largely overlapping with our slice-based quantification of C-FOS signals. For both genotypes, we included 1 additional brain sample for each that had mistargeted virus expression in the control condition. Behavior assays are based on $n = 7$ WT and $n = 7$ *Tbr1*$^{+/-}$ mice with the correct virus expression and optic fiber insertion in the BLA. The exact n numbers of experimental groups for every analysis can be found in the figure legends. For the data distribution, we did not use statistical methods to test data normality but we assumed most of the quantified data is not normally distributed. Therefore, nonparametric tests were used for statistical analysis. Statistical details can be found in the method sections describing each analysis. Data representing statistical parameters such as mean ± SEM are described in the figure legends. The test statistics (e.g., exact *P* value, Friedman chi-squared statistic, and degrees of freedom) of structure-wise comparisons can be found in **S1 Data,** and the test statistics of behavior analyses can be found in the figure legends (**Figs 8**, **S9,** and **S10**).

## Supporting information

**S1 Fig. Workflow for image segmentation of oChIEF and C-FOS signals.**
(JPG)

**S2 Fig. Spatial distribution of oChIEF-Citrine-expressing AAV injection sites in WT and Tbr1**$^{+/-}$ **mouse brains.**
(JPG)

**S3 Fig. Slice-based quantification of oChIEF and C-FOS signals and subsequent analyses.**
(JPG)

**S4 Fig. Volume-based quantification of C-FOS signals for network analysis based on the C-FOS correlations.**
(JPG)

**S5 Fig. Degree centrality (DC) of C-FOS correlation networks.**
(JPG)

**S6 Fig. Parameter tuning for community detection and minor communities of C-FOS correlation networks of the WT ctrl and WT TBS groups.**
(JPG)

**S7 Fig. Summary of the region compositions in the communities of C-FOS correlation networks.**
(JPG)

**S8 Fig. Comparison of BLA-derived circuits and C-FOS**$^+$ **cell distributions in the brain regions belonging to the default mode network (DMN).**
(JPG)

**S9 Fig. Comparisons of nonsocial behaviors during the pre- and posttreatment sessions.**
(JPG)

**S10 Fig. The effect of TBS at BLA on nonsocial behaviors prior to a reciprocal social interaction test.**
(JPG)

**S1 Data. The results of oChIEF-Citrine and C-FOS signals using slice-based analyses (related to Figs 2, 3, and S8). N/A, not available; the signals were too low to be subjected to statistical analysis.**
(XLSX)

**S2 Data. The numerical data of correlations of BLA axonal projection patterns and C-FOS expression (related to Fig 4).**
(XLSX)

**S3 Data. The numerical data of interregional correlations of C-FOS expression (related to Figs 5–7 and S5–S7).**
(XLSX)

**S4 Data. The numerical data of behavioral tests (related to Figs 8, S9, and S10).**
(XLSX)

**S1 Video.** Coronal brain slice series of binarized oChIEF signals (green) superimposed on a CCFv3 template (gray). WT (WT), upper; *Tbr1*$^{+/-}$, lower.
(M4V)

**S2 Video.** Summarized 3D stacks with binarized and binned (10 μm$^3$) oChIEF signals of WT (upper) and *Tbr1*$^{+/-}$ (lower) mouse brains.
(M4V)

## Acknowledgments

We thank Allen Institute (http://portal.brain-map.org) for the open resource about the mouse brain atlas, the Imaging Core and Animal Facility of the Institute of Molecular Biology, Academia Sinica for technical assistance, Dr. John O'Brien for English editing, and members of Y.-P. H.'s laboratory for technical assistance and discussion.

## Author Contributions

**Conceptualization:** Tsan-Ting Hsu, Yi-Ping Hsueh.

**Funding acquisition:** Yi-Ping Hsueh.

**Investigation:** Tsan-Ting Hsu, Tzyy-Nan Huang.

**Methodology:** Tsan-Ting Hsu, Tzyy-Nan Huang.

**Project administration:** Yi-Ping Hsueh.

**Software:** Tsan-Ting Hsu, Chien-Yao Wang.

**Supervision:** Yi-Ping Hsueh.

**Writing – original draft:** Tsan-Ting Hsu, Yi-Ping Hsueh.

**Writing – review & editing:** Tsan-Ting Hsu, Tzyy-Nan Huang, Chien-Yao Wang, Yi-Ping Hsueh.

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
