## [Editor Report · Decision Letter 0]

26 Apr 2024

Dear Yi-Ping, 

Thank you for submitting your manuscript entitled "Deep brain stimulation at the basolateral amygdala alters whole-brain connectivity and synchronization and social behaviors of Tbr1 deficient mice" for consideration as a Research Article by PLOS Biology.

Your manuscript has now been evaluated by the PLOS Biology editorial staff and the Academic Editor and I am writing to let you know that we would like to send your revised manuscript back to peer review. 

Please note, that given the shift in focus and new data provided in the revision, we feel that additional expertise is needed to assess the revision. Therefore, we are likely to invite one additional reviewer to assess the revised manuscript.

Once your full submission is complete, your paper will undergo a series of checks in preparation for peer review. After your manuscript has passed the checks it will be sent out for review. To provide the metadata for your submission, please Login to Editorial Manager (https://www.editorialmanager.com/pbiology) within two working days, i.e. by Apr 28 2024 11:59PM.

Kind regards,

Christian

Christian Schnell, PhD

Senior Editor

PLOS Biology

cschnell@plos.org

---

## [Decision Letter · Decision Letter 1]

24 May 2024

Dear Yi-Ping,

Thank you for your patience while we considered your revised manuscript "Deep brain stimulation at the basolateral amygdala alters whole-brain connectivity and synchronization and social behaviors of Tbr1 deficient mice" for consideration as a Research Article at PLOS Biology. Your revised study has now been evaluated by the PLOS Biology editors, the Academic Editor, the original reviewers, and one additional reviewer to evaluate the newly added data. 

In light of the reviews, which you will find at the end of this email, we are pleased to offer you the opportunity to address the remaining points from the reviewers in a revision that we anticipate should not take you very long and likely does not require the inclusion of new experimental data. We will then assess your revised manuscript and your response to the reviewers' comments with our Academic Editor aiming to avoid further rounds of peer-review, although might need to consult with the reviewers, depending on the nature of the revisions.

**IMPORTANT - SUBMITTING YOUR REVISION**

*Resubmission Checklist*

*Published Peer Review*

*PLOS Data Policy*

*Blot and Gel Data Policy*

Sincerely,

Christian

Christian Schnell, PhD

Senior Editor

PLOS Biology

cschnell@plos.org

REVIEWS:

Reviewer's Responses to Questions

Reviewer #1: This manuscript investigates impaired structural and functional connectivity, aberrant whole-brain synchronization, and the impacts of deep brain stimulation at the amygdala on whole-brain synchronization and social behaviors in a Tbr1+/- autism mouse model. The authors first established a powerful whole-brain analytical platform for immunostaining and imaging, enabling the visualization and quantification of structural and functional deficits, particularly aberrant whole-brain synchronization, caused by Tbr1 haploinsufficiency. The authors further investigate the potential therapeutic implications of deep brain stimulation at the amygdala for TBR1-linked autism. They analyze how theta-burst stimulation-mediated basolateral amygdala activation may reverse the defective synchronization at a whole-brain scale caused by Tbr1 deficiency, leading to improvements in social behaviors. Their findings suggest that BLA-derived circuits play a critical role in Tbr1-linked ASD phenotypes, highlighting the therapeutic potential of deep brain stimulation at the amygdala. Overall, the manuscript presents a stagger amount of data about the neural circuitry and network alterations underlying TBR1-linked ASD, offering novel insights into the potential therapeutic implications of deep brain stimulation at the amygdala.

I have one suggestion to improve the manuscript. It would be beneficial for the authors to discuss the duration of the behavioral changes induced by theta-burst stimulation at the basolateral amygdala and possible mechanisms underlying these changes.

Reviewer #2: The manuscript applied a whole-brain immunostaining and quantification platform to demonstrate impaired structural and functional connectivity and aberrant whole-brain synchronization in a Tbr1+/- autism mouse model. Their results emphasized the defective synchronization at a whole-brain scale caused by Tbr1 deficiency and implies a potential beneficial effect of deep brain stimulation at the amygdala for TBR1-linked autism. In addition, the current study provided the first evidence linking the DMN with the amygdala in a mouse genetic model. 

Major comments:

1 The authors summed C-FOS signals across the entire LA and BLA, only the LA exhibited a difference between WT TBS and WT Ctrl. For the BLA, statistical analysis did not reveal a difference. While all injection sites were based on the Allen Mouse Common Coordinate Framework version 3 (CCFv3), the AAV injection sites covered the LA, BLAa and BLAp, which represent BLA in the previous publication (Nat Neurosci. 2014 Feb;17(2):240-7.). The C-FOS signals in BLA did not reveal a difference, the deep brain stimulation at the BLA can lead to partial improvement in social behavior. How can this result be explained? In addition, this study used unilateral stimuli for TBS. So, will bilateral stimulation enhance the effect of behavioral improvement in autism? 

2 Figure S2 showed Spatial distribution of oChIEF-Citrine-expressing AAV injection sites in WT and Tbr1+/- mouse brains. Does the difference in injection sites affect the results of BLA circuit rewiring? 

3 The authors conducted the network analysis of functional connectivity based on inter-regional C-FOS correlations. The degree centrality (DC) and participation coefficient (PC) of individual nodes (brain regions) were calculated using NetworkX. Was this network data analysis related to real connectivity data between brain regions (according to previous literature reports), in addition to inter regional C-FOS correlations? 

4 BLA also plays a very important role in many physiological processes, such as learning and memory. The cell types of BLA are also constantly being analyzed. So, should we consider using theta burst stimulation (TBS) to regulate the specific type of neuron in BLA?

5 TBR1, a marker of glutamatergic projection neurons in the forebrain, is specifically expressed in the cerebral cortex, amygdala, olfactory bulb and hippocampus. The results showed that Tbr1+/- mice generally had more C-FOS+ cells than WT mice without any specific stimulation. Does this mean that in some brain regions, symptoms of autism can be treated by inhibiting neuronal excitation? 

Reviewer #3: The authors have significantly improved the presentation of their results. Much effort has been put into the figures. Some of the figures have been significantly modified, and the presentation of the results is now much clearer. Finally, there is a new data set on the behavioral analysis of batches of mice before and after TBS. This latest set of experiments provides a concrete measure of the impact of TBS on animal behavior and, importantly, demonstrates the positive role of TBS on the social behavior of Tbr1+/- animals. 

All the changes made to this article are responding to the original comments (although I haven't found a point-by-point response letter) and even go beyond the original requests with the addition of behavioral analysis. 

Minor comment: I am surprised that the striatum is not among the regions listed given the severe neocortical defects observed in Tbr1-/- mutants. Indeed, cortico-striatal connections are increased in these animals, and I would have expected the striatum to be similarly affected in Tbr1+/- heterozygous animals.

---

## [Editor Report · Decision Letter 2]

13 Jun 2024

Dear Yi-Ping,

Thank you for your patience while we considered your revised manuscript "Deep brain stimulation at the basolateral amygdala alters whole-brain connectivity and synchronization and social behaviors of Tbr1 deficient mice" for publication as a Research Article at PLOS Biology. This revised version of your manuscript has been evaluated by the PLOS Biology editors and the Academic Editor.

Based on our Academic Editor's assessment of your revision, we are likely to accept this manuscript for publication, provided you satisfactorily address the following data and other policy-related requests.

* We would like to suggest a different title to improve readability/accuracy: "Deep brain stimulation of the Tbr1-deficient mouse model of autism spectrum disorder at the basolateral amygdala alters amygdalar connectivity, whole-brain synchronization and social behaviors"

* DATA POLICY:

Regardless of the method selected, please ensure that you provide the individual numerical values that underlie the summary data displayed in the following figure panels as they are essential for readers to assess your analysis and to reproduce it: 8BC, S9 and S1.

* CODE POLICY

Per journal policy, if you have generated any custom code during the course of this investigation, please make it available without restrictions. Please ensure that the code is sufficiently well documented and reusable, and that your Data Statement in the Editorial Manager submission system accurately describes where your code can be found. The current link to the github repository does not work.

Please also note that we cannot accept sole deposition of code in GitHub, as this could be changed after publication. However, you can archive this version of your publicly available GitHub code to Zenodo. Once you do this, it will generate a DOI number, which you will need to provide in the Data Accessibility Statement (you are welcome to also provide the GitHub access information). See the process for doing this here: https://docs.github.com/en/repositories/archiving-a-github-repository/referencing-and-citing-content

We expect to receive your revised manuscript within two weeks. 

*Published Peer Review History*

*Press*

Sincerely,

Christian

Christian Schnell, PhD

Senior Editor

cschnell@plos.org

PLOS Biology

---

## [Editor Report · Decision Letter 3]

21 Jun 2024

Dear Yi-Ping,

Thank you for the submission of your revised Research Article "Deep brain stimulation of the Tbr1-deficient mouse model of autism spectrum disorder at the basolateral amygdala alters amygdalar connectivity, whole-brain synchronization and social behaviors" for publication in PLOS Biology. On behalf of my colleagues and the Academic Editor, J. Julius Zhu, I am pleased to say that we can in principle accept your manuscript for publication, provided you address any remaining formatting and reporting issues. These will be detailed in an email you should receive within 2-3 business days from our colleagues in the journal operations team; no action is required from you until then. Please note that we will not be able to formally accept your manuscript and schedule it for publication until you have completed any requested changes.

PRESS

Sincerely, 

Christian

Christian Schnell, PhD

Senior Editor

PLOS Biology

cschnell@plos.org